# Evaluating reanalysis representations of climatological trace gas distributions in the Asian monsoon tropopause layer

Jonathon S. Wright[1], Shenglong Zhang[1,2], Jiao Chen[1], Sean M. Davis[3], Paul Konopka[4], Mengqian Lu[2], Xiaolu Yan[5], and Guang J. Zhang[6]

[1]Department of Earth System Science, Tsinghua University, Beijing, China
[2]Department of Civil and Environmental Engineering, The Hong Kong University of Science and Technology, Hong Kong, China
[3]Earth System Research Laboratory, National Oceanic and Atmospheric Administration, Boulder, Colorado, USA
[4]Forschungszentrum Jülich (IEK-7: Stratosphere), Jülich, Germany
[5]Institute of Tibetan Plateau Meteorology, Chinese Academy of Meteorological Sciences, Beijing, China
[6]Scripps Institution of Oceanography, La Jolla, California, USA

**Correspondence:** Jonathon S. Wright (jswright@tsinghua.edu.cn)

**Abstract.** Trace gas distributions in the upper troposphere and lower stratosphere (UTLS) have important radiative and chemical impacts on climate. Although researchers have traditionally shunned direct outputs from reanalysis products at these altitudes, a looming gap in satellite observations may soon render these products essential. Here we use data from the Aura Microwave Limb Sounder (MLS) and five meteorological and composition-focused reanalyses to address two questions: Can current reanalyses reproduce essential features of UTLS composition above the Asian summer monsoon (ASM)? If so, do they reproduce these distributions from internal physics and dynamics or depend on data assimilation? All evaluated reanalyses capture regional water vapor anomalies despite moist biases in the zonal mean. Reanalysis water vapor budgets reveal the expected balance between advective hydration and 'cold trap' dehydration near the cold point; however, data assimilation effects are also influential. The scientific utility of reanalysis water vapor fields at these altitudes could be enhanced by suppressing assimilation effects to facilitate the dominant 'advection–condensation' balance, as is now done by ECMWF. The two reanalyses that provide CO show good agreement with observed convective enhancement, highlighting the value of including CO-like transport tracers in reanalyses. All five reanalyses also reproduce the seasonal 'ozone valley' above the monsoon reasonably well, but the only reanalysis to provide a complete ozone budget relies heavily on data assimilation to do so. The composition reanalyses, with more sophisticated chemistry, provide a better match to ozone observations, but it remains unclear whether they can do so without Aura MLS.

## 1 Introduction

During boreal summer, anomalies in the upper-level anticyclone above the Asian summer monsoon (ASM) dominate circulation and composition fields near the tropopause (e.g. Krishnamurti, 1971; Park et al., 2007; Santee et al., 2017). This upper-level anticyclone, which is generated by diabatic heating associated with ASM convection and the Tibetan Plateau (Hoskins and Rodwell, 1995; Garny and Randel, 2013; Liu et al., 2017; Siu and Bowman, 2019), spans nearly a third of the Northern

Hemisphere from the Middle East to the western Pacific. Vertically, the depth of the anticyclone extends from the base of the tropical easterly jet ($\sim$200 hPa near 15°N) to the top of the subtropical westerly jet ($\sim$70 hPa near 40°N; Randel and Park, 2006; Park et al., 2007). Mixing and transport along the boundaries of the anticyclone transmit the properties of anticyclone air to all corners of the globe (Dethof et al., 1999; Popovic and Plumb, 2001; Ploeger et al., 2013; Yu et al., 2017; Yan et al., 2019; Honomichl and Pan, 2020; Pan et al., 2022).

Despite ample evidence of strong coupling among convection, clouds, composition, and circulation in this region, the net balance and effects of these processes remain elusive. ASM deep convection, which typically begins in April and matures in July and August (Webster et al., 1998; Romatschke et al., 2010; Qie et al., 2014), penetrates upward to potential temperatures of 360 K and above (Fu et al., 2006; Luo et al., 2011; Qie et al., 2014; Vogel et al., 2019; Bucci et al., 2020). These deep convective systems influence the structure of the tropopause layer (Kumar et al., 2015, 2018; Muhsin et al., 2018) and contribute to the relatively humid lower stratosphere above the ASM (e.g. Fu et al., 2006; Wright et al., 2011; Tissier and Legras, 2016; Ueyama et al., 2018; Singh et al., 2021). Convection also leads to low concentrations of ozone and high concentrations of CO and other pollutants in the upper troposphere and lower stratosphere (UTLS) relative to other locations at the same latitude (Park et al., 2007; Pan et al., 2016; Santee et al., 2017; Kumar and Ratnam, 2021; Gao et al., 2023). These regional anomalies in ozone and CO are then modified by local chemical processes and transport (Randel and Park, 2006; Gottschaldt et al., 2018; von Hobe et al., 2021). Global climate models struggle to produce consistent representations of the anticyclone and exclude many relevant cloud and chemical processes in the region, such as those related to aerosol chemistry and microphysics (Xue et al., 2017; Singh et al., 2022). These model deficiencies are compounded by the limited accuracy, spatial coverage, and temporal sampling frequencies of observations targeting this layer (e.g. Hegglin et al., 2013; Khosrawi et al., 2018), as well as inconsistencies in key aspects of the anticyclone in atmospheric reanalyses (Nützel et al., 2016; Manney et al., 2021).

Early reanalysis systems were unable to reproduce variations in water vapor and other key constituents, particularly at UTLS altitudes (Davis et al., 2017, and references therein). As a result, researchers have typically used reanalysis winds and temperatures to drive chemical transport or Lagrangian trajectory models when exploring the processes that control water vapor and other constituents at and above the tropopause (e.g. Fueglistaler et al., 2005; Konopka et al., 2023). Such simulations often adopt the 'advection–condensation' paradigm, in which condensation is assumed to occur at a fixed relative humidity and condensates are removed immediately (e.g. Liu et al., 2010). However, Lagrangian simulations may differ substantially when the same model is driven by different reanalysis products (e.g. Wright et al., 2011; Schoeberl et al., 2012; Tao et al., 2019).

In a recent evaluation of zonal-mean ozone and water vapor in reanalyses, users of these products were advised to proceed with caution in the UTLS due to biases in both species on the order of $\sim$50% (Davis et al., 2022). In addition, studies based on these products should demonstrate that their results are not sensitive to the choice of reanalysis and, where possible, check results based on reanalysis trace gas products against independent observations. Despite these concerns, each individual reanalysis provides a comprehensive perspective on both the atmospheric state and the processes involved in its evolution. Moreover, most recent reanalyses provide budget and tendency terms alongside core meteorological fields, presenting new opportunities to evaluate reanalysis atmospheric composition fields and the mechanisms behind their evolution. Reanalysis centers are also increasingly providing access to model-generated forecast variables (i.e. the background state for data assimilation), allowing

us to compute assimilation increments that measure differences between the reanalysis state before and after data assimilation. These increments may be especially illuminating for the UTLS, where different reanalyses apply very different constraints on the influences of assimilated observations (Davis et al., 2022). Careful evaluation and intercomparison of reanalysis products can therefore provide multiple perspectives on the mechanisms governing UTLS composition in the real atmosphere (Fujiwara et al., 2022).

In this study, we investigate the climatological distributions of UTLS water vapor and chemical tracers above the ASM in the context of dynamical and thermodynamic fields from reanalyses. We conduct this analysis using satellite data, primarily from the Aura Microwave Limb Sounder (MLS) and other instruments from the A-Train constellation of satellites (LEcuyer and Jiang, 2010), along with products from five recent atmospheric reanalyses (see Sect. 2.1). This study has two objectives. First, to evaluate the performance of current reanalysis products in reproducing the climatology and seasonal cycle of water vapor and other trace gases above the ASM. Second, to quantify the roles of advection, parameterized physical processes, and data assimilation in producing those climatological distributions. Further evaluation of the mechanisms and modes of variability governing subseasonal-to-interannual fluctuations in reanalysis water vapor and trace gas products in the monsoon UTLS is provided in a follow-up to this paper (Zhang et al., 2025). This work updates and extends results presented in Chapter 8 of the Stratosphere-troposphere Process and their Role in Climate (SPARC) Reanalysis Intercomparison Project (S-RIP) Final Report (Fujiwara et al., 2022; Tegtmeier et al., 2022).

The remainder of this paper is organized as follows. In Sect. 2, we introduce the data and methods used in the analysis, along with key details of the reanalysis products. In Sect. 3, we describe the climatological distributions and seasonal cycles of water vapor, ozone, and CO above the ASM and differences relative to the distributions retrieved by Aura MLS. In Sect. 4, we evaluate reanalysis representations of dynamical, physical, and data assimilation influences on water vapor and ozone in the ASM tropopause layer. We conclude with a brief summary and outlook in Sect. 5.

## 2 Data and method

### 2.1 Reanalysis products

We use model-level and pressure-level outputs from five recent atmospheric reanalyses (Table 1). These include the European Centre for Medium-Range Weather Forecasts (ECMWF) Fifth Reanalysis of the Atmosphere (ERA5; Hersbach et al., 2020), the Japanese Reanalysis for Three Quarters of a Century (JRA-3Q; Kosaka et al., 2024), the Modern-Era Retrospective Analysis for Research and Applications, version 2 (MERRA-2; Gelaro et al., 2017), the MERRA-2 Stratospheric Composition Reanalysis of Aura Microwave Limb Sounder (M2-SCREAM; Wargan et al., 2023), and the Copernicus Atmosphere Monitoring Service (CAMS) reanalysis of atmospheric composition (Inness et al., 2019). These five reanalysis products represent three different forecast models and five different data assimilation systems, although some of the products are closely interrelated (Table 1). The vertical resolutions of the products we use are shown in Fig. 1.

ERA5 and CAMS are based on essentially the same forecast model and meteorological data assimilation system, although CAMS is conducted at a coarser resolution ($\sim$79 km and 60 vertical levels) compared to ERA5 ($\sim$31 km and 137 vertical

**Table 1.** Basic information on the five atmospheric reanalyses evaluated in this work: name, reanalysis centre, data assimilation system, horizontal grid, and vertical grid (number of levels and model top), frequency of assimilated fields, and reference.

| Name | Centre | Data Assimilation System | Grid | Levels | Frequency | Reference |
|------|--------|--------------------------|------|--------|-----------|-----------|
| CAMS | ECMWF | IFS Cy42r1 4DVar | $T_L 255$ (∼79 km) | 60 to 10 Pa | 3 h | Inness et al. (2019) |
| ERA5 | ECMWF | IFS Cy41r2 4DVar | $T_L 639$ (∼31 km) | 137 to 1 Pa | 1 h | Hersbach et al. (2020) |
| JRA-3Q | JMA | JMA 4DVar | $T_L 479$ (∼40 km) | 100 to 1 Pa | 6 h | Kosaka et al. (2024) |
| MERRA-2 | NASA GMAO | GEOS-5.12.4 IAU | $0.625° \times 0.5°$ | 72 to 1 Pa | 3 h | Gelaro et al. (2017) |
| M2-SCREAM | NASA GMAO | GEOS CoDAS + replay | $0.625° \times 0.5°$ | 72 to 1 Pa | 3 h | Wargan et al. (2023) |

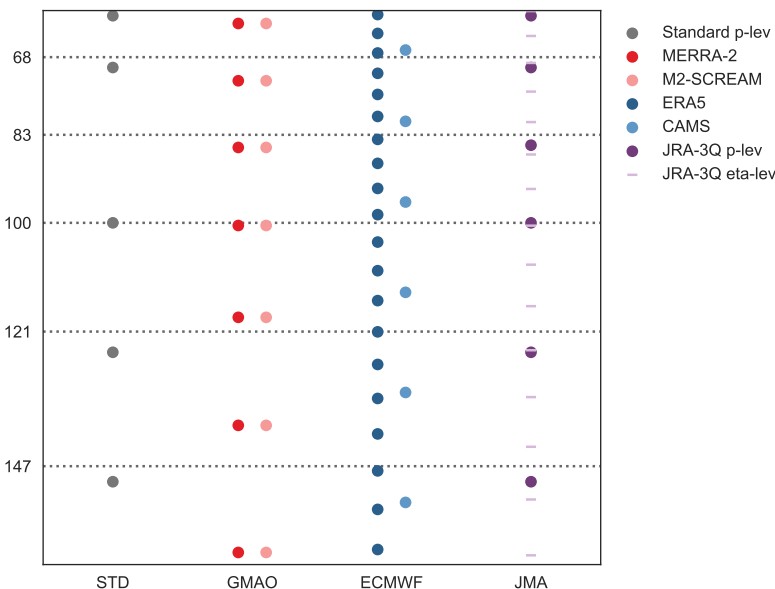

**Figure 1.** Vertical levels based on a surface pressure of 1000 hPa for MERRA-2 (dark red), M2-SCREAM (light red), ERA5 (dark blue), CAMS (light blue), and JRA-3Q (purple). The analysis presented in the manuscript is based on pressure-level fields for JRA-3Q (purple circles); locations of model levels are shown as light purple dashes for context. Other reanalyses provide pressure-level products on the standard pressure-level grid (grey circles), which has no counterpart to the 83 hPa Aura MLS level. All reanalysis products are interpolated to the Aura MLS pressure levels marked by horizontal dotted lines.

levels). In addition to model resolution, CAMS differs from ERA5 in its use of analyzed ozone rather than prescribed ozone for radiation calculations (Inness et al., 2019; Hersbach et al., 2020). The M2-SCREAM reanalysis is based on a 'replay' (Orbe et al., 2017) of MERRA-2 meteorological fields and is therefore similar to MERRA-2 in winds and temperatures while differing substantially in the composition of the upper troposphere and stratosphere (Wargan et al., 2023). For example, aided by assimilated Aura MLS retrievals, M2-SCREAM represents spatiotemporal variations of water vapor throughout the strato-

sphere (216–0.01 hPa; Wargan et al., 2023). By contrast, MERRA-2 relaxes water vapor to a climatology, damping virtually all variability at pressures less than 50 hPa and some variability between 50 hPa and the tropopause (Davis et al., 2017; Gelaro et al., 2017). Moreover, whereas ERA5, MERRA-2, and JRA-3Q are primarily meteorological reanalyses that include some information on atmospheric composition, CAMS and M2-SCREAM focus mainly on atmospheric composition. Among the two composition-focused reanalyses, CAMS is more oriented toward tropospheric composition, while M2-SCREAM specifically targets the upper troposphere and stratosphere. This diversity among the selected products is intended to provide additional context to both the MLS-based results and the reanalysis intercomparison. JRA-3Q is the only reanalysis that does not assimilate ozone retrievals from Aura MLS, while M2-SCREAM is the only reanalysis to assimilate Aura MLS retrievals of water vapor (Table 2). These distinctions affect the independence of comparisons between Aura MLS and the reanalysis products and thus our interpretation of the results. None of the reanalyses assimilate CO retrievals from Aura MLS.

We use several other reanalysis-based variables to describe the state of the monsoon tropopause layer. First, we use specific humidities, ozone mass mixing ratios, and CO mass mixing ratios from different reanalysis products. Among the reanalyses, only CAMS and MERRA-2 provide estimates of CO and only CAMS includes any CO-related data assimilation (Table 2). Second, we use Montgomery streamfunction (MSF $= c_p T + \Phi$, where $c_p$ is the specific heat of dry air at constant pressure, $T$ is temperature, and $\Phi$ is geopotential) on the 395 K isentropic surface to define the geographic boundaries of the anticyclone (Santee et al., 2017; Manney et al., 2021). MSF, which is functionally equivalent to dry static energy, defines the geostrophic wind on an isentropic surface and is thus analogous to geopotential in pressure coordinates. The 395 K isentropic surface is typically located between 100 hPa and 83 hPa in the core ASM region (i.e. south of the subtropical westerly jet). Third, we use horizontal winds, temperatures, the level of zero net radiative heating (LZRH), and cold point tropopause (CPT) temperatures (identified following Tegtmeier et al., 2020) from each reanalysis to evaluate the climatological transport environment within the ASM UTLS. Winds and temperatures describe the structure of the anticyclone. The LZRH marks the typical boundary between subsidence balanced by radiative cooling in the upper troposphere and ascent balanced by radiative heating in the tropopause layer and lower stratosphere. The CPT temperature provides a measure of the minimum temperatures in the tropopause layer, which control dehydration of water vapor in ascending air parcels. Finally, we use tendency terms from ERA5, JRA-3Q, and MERRA-2 to analyze time-mean water vapor and ozone budgets.

The vertically-resolved water vapor budget can be expressed as (e.g. Peixoto and Oort, 1992, their eq. 12.6):

$$\frac{\partial q}{\partial t} + \nabla \cdot (\mathbf{V} q) + \frac{\partial (\omega q)}{\partial p} = S \tag{1}$$

where $q$ is specific humidity, $p$ is pressure, $\mathbf{V}$ is the vector horizontal wind on an isobaric surface, and $\omega$ is the pressure vertical velocity. The terms on the left-hand side represent the time rate of change in specific humidity, the horizontal moisture flux divergence, and the vertical moisture flux divergence, respectively. The term $S$ on the right-hand side represents local sources and sinks. We compute the moisture flux divergence terms hourly on a $1°\times1°$ latitude–longitude grid for ERA5 and 6-hourly on a $1.25°\times1.25°$ latitude–longitude grid for JRA-3Q using the windspharm package (Dawson, 2016). A T42 spectral filter is applied to time-mean budget terms to reduce noise in the spatial patterns. This filter simplifies the visualizations and does not change the results in any significant way.

In our budget, the source-sink term $S$ comprises the influences of parameterized physical processes (cloud microphysics, convection, turbulent mixing, etc.), data assimilation, and a residual:

$$S = S_{\mathrm{phy}} + S_{\mathrm{ana}} + S_{\mathrm{res}} \qquad (2)$$

In the following, we refer to the data assimilation term $S_{\mathrm{ana}}$ as the assimilation increment. Many reanalyses, including ERA5, JRA-3Q, and MERRA-2, provide $S_{\mathrm{phy}}$ from the forecast model as diagnostic output. MERRA-2 also explicitly provides $S_{\mathrm{ana}}$, but this component must be estimated for budgets based on ERA5 and JRA-3Q. We estimate analysis increments ($S_{\mathrm{ana}}$) for ERA5 and JRA-3Q by directly subtracting forecast specific humidities (based on the model background state before data assimilation) from analysis specific humidities (the reanalysis state after data assimilation). To convert this into a rate, we then average over all time steps and multiply by the number of analysis cycles per day (two for ERA5 and four for JRA-3Q) to yield assimilation-related moistening per day. This approach is equivalent to the analysis increment diagnostic provided by MERRA-2, which is computed at the end of the 'predictor' step and applied over the subsequent 'corrector' step in the incremental analysis update data assimilation system used by MERRA-2 (Gelaro et al., 2017; Fujiwara et al., 2017). The residual ($S_{\mathrm{res}}$) collects the effects of numerical diffusion as well as high-frequency transport terms. The latter contributions to this term arise for two reasons. First, the forecast model horizontal resolution is finer ($\sim$31 km for ERA5; $\sim$40 km for JRA-3Q) than the fields we use to calculate $S_{\mathrm{dyn}}$ (1° and 1.25°, respectively). Second, the forecast model time step (12 minutes for both systems) is much finer than the analysis interval (1 h for ERA5 and 6 h for JRA-3Q). Transports on space or time scales between these resolutions are omitted from $S_{\mathrm{dyn}}$ because our calculation cannot resolve them and from $S_{\mathrm{phy}}$ because the model does resolve them. See Fig. 10 and related text for further details.

## 2.2 Satellite data

The Microwave Limb Sounder (MLS; Waters et al., 2006) on board the Aura satellite has measured water vapor and other trace gases in the upper troposphere and stratosphere ($p < 316$ hPa) since August 2004. Aura MLS has been shown to reliably capture temporal, vertical, and horizontal variations in water vapor and other trace gases in the Asian monsoon region (Yan et al., 2016; Santee et al., 2017). We use MLS version 5 (v5) gridded retrievals of water vapor (Lambert et al., 2021), ozone (Schwartz et al., 2021a), carbon monoxide (Schwartz et al., 2021c), and temperature (Schwartz et al., 2021b) from 2005–2021 covering the region 30°E–130°E and 15°N–45°N. Whereas water vapor and ozone are provided on five levels within our analysis domain (147 hPa, 121 hPa, 100 hPa, 83 hPa, and 68 hPa; Fig. 1), CO is provided on every other level (147 hPa, 100 hPa, and 68 hPa). Compared to MLS v4.2 (Livesey et al., 2017), the "slow drift" issue in water vapor retrievals (Hurst et al., 2016) has been improved in v5 (Livesey et al., 2021, 2022), while v5 ozone data shows negligible differences from v4 in the stratosphere (Livesey et al., 2021, 2022; Stauffer et al., 2022). The climatological mean location of the lapse-rate tropopause (based on the World Meteorological Organization definition; e.g. Hoffmann and Spang, 2022) is provided based on Level 3 gridded monthly-mean data at 1°×1° resolution from the Atmospheric Infrared Sounder (AIRS), version 7 (Aumann et al., 2003; Kahn et al., 2023). The climatological mean distribution of outgoing longwave radiation (OLR) over the warm seasons of 2005–2021 is taken

**Table 2.** Chemistry and stratospheric ozone schemes along with data assimilated by each reanalysis system for upper tropospheric and lower stratospheric water vapor, ozone, and carbon monoxide (CO) during the analysis period (2005–2021).

| Name | Chemistry | Ozone Model | Water vapor | Ozone | CO |
|---|---|---|---|---|---|
| CAMS | IFS(CB05)[a] | Cariolle[b] | Up to tropopause[c] | OMI, SCIAMACHY, SBUV/2, MLS, GOME-2, MIPAS | MOPITT |
| ERA5 | None | Cariolle[b] | Up to tropopause[c] | OMI, SCIAMACHY, SBUV/2, MLS, GOME-2, MIPAS | None |
| JRA-3Q | None | MRI-CCM2.1 | $T \geq -40°C$[d] | Naoe et al. (2020) | None |
| MERRA-2 | None | PCHEM[e] | Up to $300\,\mathrm{hPa}$[d] | SBUV, MLS, OMI | Model[f] |
| M2-SCREAM | StratChem[g] | StratChem | MLS | MLS, OMI | None |

[a] Tropospheric chemistry as described by Flemming et al. (2015); see also Inness et al. (2019)

[b] Linearized ozone parameterization (Cariolle and Teyssèdre, 2007; Dragani, 2011)

[c] Observations are assimilated up to this level; no assimilation increments are allowed above this level

[d] Restriction applies to direct measurements; satellite radiances or radio occultation bending angles are allowed and increments may influence higher levels

[e] Zonally-symmetric monthly production and loss derived from a 2-dimensional model (Nielsen et al., 2017; Wargan et al., 2017)

[f] The MERRA-2 implementation of CO omitted some emission sectors (K. Wargan, personal communication)

[g] Detailed stratospheric chemistry only as described by Nielsen et al. (2017); see also Wargan et al. (2020)

from the Clouds and the Earth Radiant Energy System (CERES) Energy Balanced and Filled (EBAF) product (Edition 4.2; Loeb et al., 2020).

## 2.3 Methodology

We evaluate distributions of water vapor and other trace gases in the tropopause layer above the ASM (147–68 hPa). This layer, which contains the cold point tropopause (CPT), is a transitional zone where air detrained from convection spirals upward into the lower stratosphere (e.g. Vogel et al., 2019). Although we retain vertical information for most evaluations, for some comparisons we integrate trace gas concentrations in pressure from $p_{\mathrm{bot}}$ (147 hPa) to $p_{\mathrm{top}}$ (68 hPa) for the target region. We refer to the integrated variable for water vapor as partial-column water vapor (PCWV) in units of mass per area (kg m$^{-2}$):

$$\mathrm{PCWV} = -\frac{1}{g} \int\limits_{p_{\mathrm{bot}}}^{p_{\mathrm{top}}} q\,dp, \tag{3}$$

where $g$ is the gravitational acceleration, $q$ is the water vapor mass mixing ratio, and $dp$ is the thickness of each isobaric layer from the layer centered at $p_{\mathrm{top}}$ (68 hPa) to the layer centered at $p_{\mathrm{bot}}$ (147 hPa). Specific humidity from each reanalysis is interpolated from model levels (or pressure levels, in the case of JRA-3Q) to the MLS pressure levels, but no weighting functions are applied. Because JRA-3Q pressure-level products include an additional level at 85 hPa (Fig. 1), these products are well matched to the Aura MLS vertical grid. MERRA-2, ERA5, and CAMS pressure-level products do not include any levels between 70 hPa and 100 hPa, necessitating the use of model-level products, while M2-SCREAM is only provided on model levels. Partial-column ozone (PCO3) and partial-column CO (PCCO) are calculated similarly from ozone and CO mass

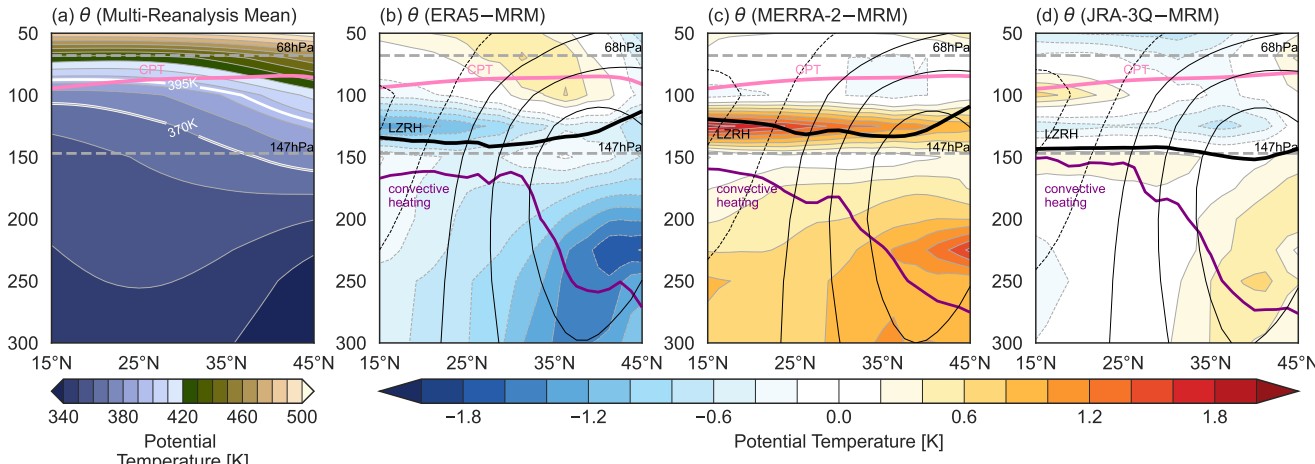

**Figure 2.** Regional-mean distributions of potential temperature ($\theta$; shading, in units of K) averaged over 40°E–110°E based on (a) the multi-reanalysis mean (MRM) including ERA5, MERRA-2, and JRA-3Q, (b) ERA5 minus the MRM, (c) MERRA-2 minus the MRM, and (d) JRA-3Q minus the MRM during May–September of 2005–2021. Locations are marked for the vertical bounds of the layer used to calculate partial-column water vapor, ozone, and CO (dashed horizontal lines), the regional-mean zonal wind (thin black contours; dotted negative; intervals of $10\,\mathrm{m\,s^{-1}}$ starting from $\pm 20\,\mathrm{m\,s^{-1}}$), the cold point tropopause (thick pink contour); the regional-mean level of zero net radiative heating (LZRH; thick black contour), and the approximate depth of zonal-mean convective heating (non-radiative heating greater than $0.3\,\mathrm{K\,d^{-1}}$; thick purple contour).

mixing ratios. We use spatial distributions of PCWV, PCO3, and PCCO to support the evaluation and intercomparison of reanalysis products in the ASM region against Aura MLS, as presented in Sect. 3.

Mean seasonal cycles are constructed by averaging daily gridded data into pentads (discrete 5-day periods) starting from 1 May and ending on 2 October and averaging across all available years. These start and end dates yield 31 pentads per year (155 days). This 'extended monsoon' analysis period is motivated by results from two recent studies examining this region (Santee et al., 2017; Manney et al., 2021). Although the anticyclone is strongest during the peak monsoon months of July and August, anomalies in the thermodynamic structure and composition of the monsoon UTLS are initiated in May as pre-monsoon convection intensifies and typically extend to the end of September when the anticyclone dissipates. Climatological distributions are constructed from the same data (i.e. 1 May – 2 October) unless indicated otherwise. We focus on the UTLS between 147 hPa and 68 hPa (approximately 370–440 K potential temperature), within the 'upward spiraling range' highlighted by Vogel et al. (2019).

## 3 Climatology and seasonal evolution

### 3.1 Thermodynamic structure of the tropopause layer

We start by establishing the dynamical and thermodynamic context of the monsoon tropopause layer. Figure 2 shows distributions of potential temperature, zonal winds, and the vertical locations of the cold point tropopause (CPT), the level of zero net radiative heating (LZRH), and the approximate depth of convective heating based on ERA5, MERRA-2, and JRA-3Q. The LZRH is plotted as the zero contour in total ($Q_{\mathrm{rad}} = Q_{\mathrm{LW}} + Q_{\mathrm{SW}}$) radiative heating, and the depth of convective heating is approximated as the $0.3\,\mathrm{K\,day^{-1}}$ contour in non-radiative ($Q_{\mathrm{non}} = Q_{\mathrm{phy}} - Q_{\mathrm{rad}}$) heating, where $Q_{\mathrm{phy}}$ is the diabatic heating due to all physics (notations follow Wright and Fueglistaler, 2013). Distributions of $Q_{\mathrm{phy}}$ based on multiple reanalyses, including ERA5 and MERRA-2 but not including JRA-3Q, are available in Chapter 8 of the S-RIP Final Report (Tegtmeier et al., 2022, their Fig. 8.59).

The distribution of potential temperature shows a warm upper troposphere (small $\partial\theta/\partial p$ within the monsoon anticyclone) and a strongly stratified lower stratosphere (large $\partial\theta/\partial p$ above; Fig. 2a). Weak stratification in the upper troposphere results from sustained convective heating that reduces $\partial\theta/\partial p$ and dilutes potential vorticity (e.g. Garny and Randel, 2013). ERA5 produces a sharper vertical gradient of potential temperature than MERRA-2 or JRA-3Q owing to its cooler upper troposphere (especially relative to MERRA-2) and warmer lower stratosphere (especially relative to JRA-3Q). Warmer temperatures in MERRA-2 relative to ERA5 (Fig. 2c) are centered in the upper troposphere (150–200 hPa) and the lower part of the tropopause layer (100–150 hPa). Differences between ERA5 and JRA-3Q are largest in the tropopause layer and lower stratosphere (50–150 hPa; Fig. 2d). The warm bias in JRA-3Q at and just above 100 hPa is especially notable because cold temperatures along the southern flank of the anticyclone regulate water vapor entering the stratosphere through this region (Fueglistaler et al., 2005; Wright et al., 2011).

The warmer upper troposphere in MERRA-2 is a known bias caused at least in part by an artificial extension of anvil cloud lifetimes aimed at improving the top-of-atmosphere radiative balance (Molod et al., 2015; Wright et al., 2020). Although these extended cloud lifetimes successfully improve agreement with observed outgoing longwave radiation (OLR), they also produce a much stronger signature in upper tropospheric cloud radiative heating than any other reanalysis (Wright et al., 2020; Tegtmeier et al., 2022). These effects alter radiative equilibrium temperatures both above and below the anvil layer, producing a warm bias that reinforces itself by increasing static stability in the upper troposphere, reducing the depth of deep convection, and promoting further detrainment of anvil ice into the stabilized layer.

The zonal-mean vertical location of the CPT matches well across the three reanalyses (see also Tegtmeier et al., 2020). Although the depth of convective heating is comparable among the three reanalyses south of 25°N, ERA5 produces deeper convective heating over 27–32°N, which includes the southern slope of the Himalayas, the southeastern Tibetan Plateau, and the Sichuan Basin (see also Legras and Bucci, 2020). Convection in this region has been argued to have an outsized influence on water vapor transport to the stratosphere via the monsoon (e.g. Fu et al., 2006; Wright et al., 2011). The zonal-mean LZRH is fairly flat and located near 150 hPa in ERA5 and JRA-3Q, with that based on ERA5 shifted to slightly higher altitudes. The zonal-mean LZRH in MERRA-2 is located even higher at around 130–135 hPa. This upward shift of the LZRH

in MERRA-2 is also a consequence of strong cloud radiative effects, namely strong cloud-top cooling and weak radiative heating in the tropopause layer above clouds (Wright et al., 2020). Warmer temperatures in the lower part of the tropopause layer (100–150 hPa; Fig. 2c) may also contribute to raising the LZRH by reducing or even reversing the difference between ambient temperatures and radiative equilibrium temperatures (Fueglistaler et al., 2009, their eq. 4 and related discussion).

Figure 3 shows vertical profiles of climatological area-mean temperature along with three-dimensional distributions of temperature in the ASM tropopause layer based on JRA-3Q, MERRA-2, M2-SCREAM, ERA5, and CAMS for 1 May–2 October of 2005–2021. Profiles based on two observational datasets are shown in Fig. 3a for reference: Aura MLS v5 temperature retrievals and profiles based on Global Navigation Satellite System radio occultation measurements from the MetOp series of satellite (Bonnedal et al., 2010). A more complete distribution of temperature based on Aura MLS is shown in Fig. S1 of the online supplement, along with the distribution of lapse-rate tropopause pressure provided by AIRS. Spatial distributions and profiles of temperature for ERA5 and the reprocessed ERA5.1 (conducted to correct a temperature bias in the UTLS; Simmons et al., 2020) are provided in Fig. S2 of the online supplement. Time-mean differences between ERA5.1 and ERA5 over the overlapping period 2005–2006 are small. We therefore use outputs from ERA5 exclusively in this work.

Radio occultation profiles have very small uncertainties in this vertical range (0.2–0.6%; Nielsen et al., 2022) and, owing to their small biases, have become valuable 'anchor' observations for calibrating bias corrections during reanalysis production (Poli et al., 2010; Wright et al., 2022). Because radio occultation bending angles influence the reanalysis temperature fields through data assimilation, their inclusion here is mainly to assess the difference between Aura MLS and the reanalyses. Although the reanalysis profiles are considerably warmer than indicated by Aura MLS in the monsoon UTLS, ERA5 and MERRA-2 are in good agreement with the radio occultation retrievals interpolated to MLS levels (dark grey line), indicating that MLS has a cold bias in this region (see also Yan et al., 2016). JRA-3Q has a roughly 0.3 K warm bias at 83 hPa relative to the radio occultation-based profile, ERA5, and MERRA-2. The higher resolution 'dry temperature' radio occultation profiles illustrate finer vertical structure on the interannual scale that is not resolved on the MLS vertical grid. Data assimilation constraints are strong on reanalysis temperature fields, yielding good overall agreement between the reanalyses. However, it is again evident that JRA-3Q simulates warmer temperatures than other reanalyses in the cold trap region over the Bay of Bengal (Fig. 3b).

## 3.2 Trace gas climatologies

Figure 4 shows the horizontal and vertical structure of water vapor anomalies in the ASM tropopause layer relative to the global (0–360°E) zonal mean based on Aura MLS, JRA-3Q, MERRA-2, M2-SCREAM, ERA5, and CAMS for 1 May–2 October of 2005–2021. See Fig. S3 in the online supplement for absolute distributions that do not subtract the zonal mean. Aura MLS shows a local maximum of water vapor in the southeastern part of the ASM anticyclone (Fig. 4a), consistent with previous results (e.g. Randel et al., 2015). The southeastern maximum in LSWV indicates moisture supplied locally by monsoon convection and isolated by the anticyclone from the drier environment outside (Legras and Bucci, 2020). This local maximum is mainly contributed by moisture near the base of the tropopause layer (∼147 hPa), with conditions slightly drier than the zonal mean both above this anomaly and to its north (35–45°N) and west (30–50°E). All five reanalyses reproduce the spatial

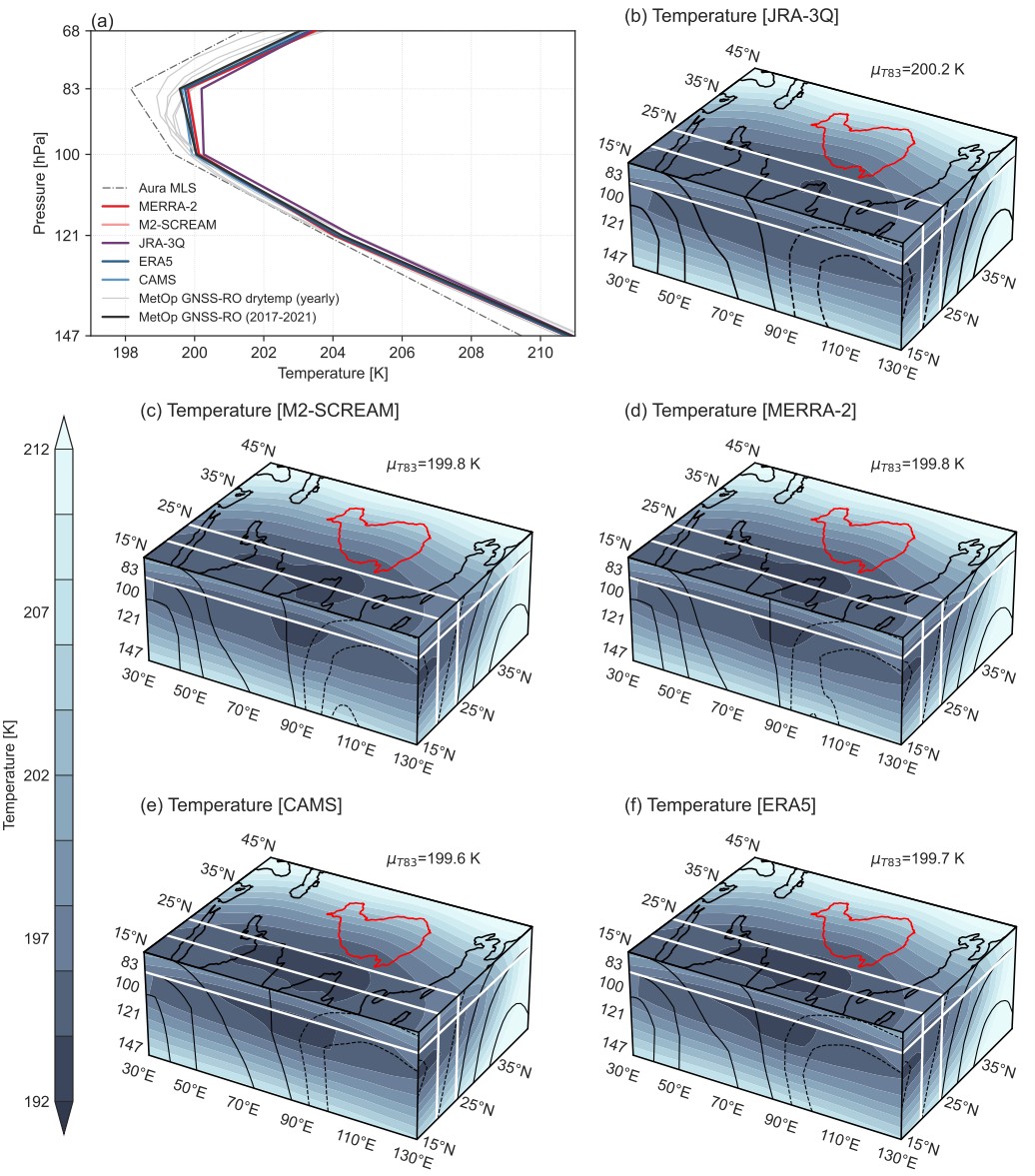

**Figure 3.** (a) Area-mean time-mean vertical profiles of temperature (units: K) based on Aura MLS, radio occultation, and the five reanalyses, along with three-dimensional distributions of temperature (units: K) based on (b) JRA-3Q, (c) M2-SCREAM, (d) MERRA-2, (e) CAMS, and (f) ERA5 during May–September 2005–2021. Top faces show temperatures on the 83 hPa isobaric surface for each reanalysis in (b)–(f), with area-mean values listed at the top right of each panel. The south (left) face in (b)–(f) shows temperature (shading) and meridional winds (contours) averaged for the east–west transect between 20°N–25°N (south face). The east (right) face in (b)–(f) shows temperature (shading) and zonal wind (contours) averaged zonally over 30°E–130°E. Red contours mark the location of the Tibetan Plateau. White lines mark the boundaries of the east–west transect and the location of the top face.

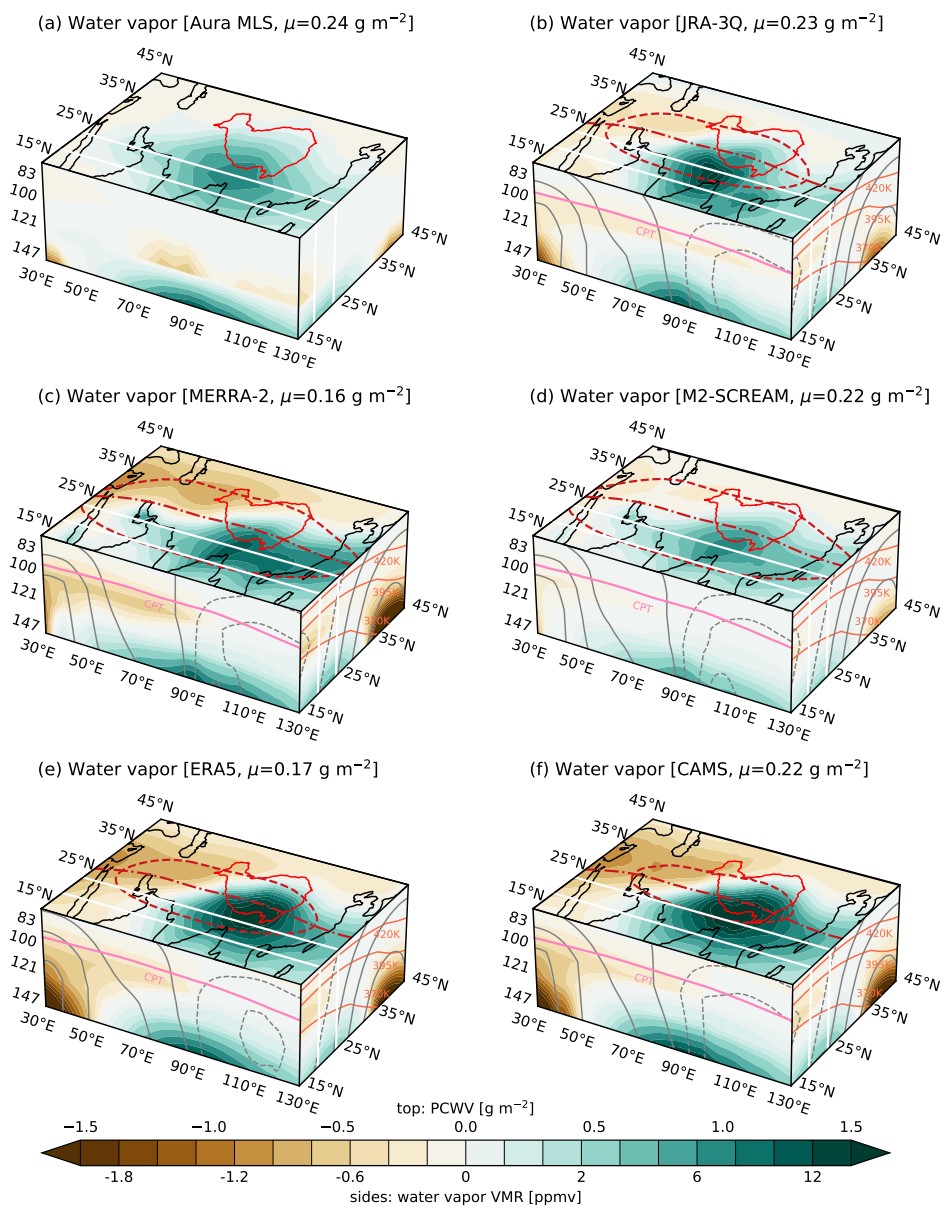

**Figure 4.** Three-dimensional structure of monsoon-region water vapor anomalies relative to the all-longitude zonal mean based on (a) Aura MLS, (b) JRA-3Q, (c) MERRA-2, (d) M2-SCREAM, (e) ERA5, and (f) CAMS during 1 May–2 October 2005–2021. The top face of each cube shows partial-column water vapor (PCWV) integrated over 68–147 hPa with area-mean values in each title, the south (left) face shows the east–west transect averaged between 20°N–25°N (white lines), and the east (right) face shows the zonal mean averaged over 30°E–130°E. The $369\,200\,\mathrm{m^2\,s^{-2}}$ contour of Montgomery streamfunction (dashed red contour) and the $0\,\mathrm{m\,s^{-1}}$ contour of zonal wind (dash-dot red contour) mark the boundary and ridgeline of the upper-level anticyclone on the 395 K isentropic surface in each reanalysis. Pink contours on the south face mark the cold-point tropopause, orange contours on the east face show potential temperature (see also Fig. 2), and grey contours show meridional (south face) and zonal (east face) winds in panels (b)–(f). Solid red contours on each top face mark the location of the Tibetan Plateau.

pattern, but with larger deviations from the zonal mean in both the humid southeast and dry northwest, possibly owing to the use of discrete levels without the deeper satellite vertical weighting functions or the elongated along-track field-of-view that characterize MLS retrievals. The vertical location of the largest dry anomalies is also shifted upward in most reanalyses relative to Aura MLS, from ∼121 hPa to just below the cold-point tropopause (∼100 hPa).

Among the reanalyses, M2-SCREAM (Fig. 4d), which assimilates an earlier version of Aura MLS water vapor (v4 rather
than v5), best reproduces the observed pattern, followed closely by JRA-3Q (Fig. 4b). The magnitude of water vapor anomalies in JRA-3Q also matches Aura MLS well, with a small moist bias of +3% in PCWV (Fig. S3a-b), comparable to that in M2-SCREAM (+4%; Fig. S3d). However, as described in sect. 3.3 below, the good agreement in PCWV between JRA-3Q and Aura MLS hides substantial compensating biases in the vertical dimension across this layer. Although the other three reanalyses, MERRA-2 (+38%), ERA5 (+24%), and CAMS (+35%), all overestimate area-mean PCWV (30°E–130°E, 15°N–
45°N; Fig. S3), both MERRA-2 (Fig. 4c) and ERA5 (Fig. 4e) underestimate regional moistening relative to the zonal mean and CAMS is within 10% of Aura MLS (Fig. 4f). The small biases in the regional anomalies based on these reanalyses indicate that moist biases relative to Aura MLS are present in the zonal mean and are not specific to the monsoon region. However, the relatively good agreement between the reanalyses and Aura MLS in the regional anomalies masks larger moist and dry anomalies and sharper spatial gradients (Fig. 4). Among the reanalyses, CAMS shows the sharpest spatial gradients
with a $2.70 \, \text{g m}^{-2}$ difference between the minimum and maximum anomalies, followed by ERA5 ($2.55 \, \text{g m}^{-2}$). M2-SCREAM ($1.29 \, \text{g m}^{-2}$) is again in best agreement with Aura MLS ($1.25 \, \text{g m}^{-2}$). These results, and particularly the zonal-mean biases, reinforce earlier conclusions that tropopause-layer water vapor is difficult for reanalyses to reproduce (Davis et al., 2017). However, they also indicate that recent reanalyses capture seasonal-mean anomalies well in this region despite the overall lack of observational constraints on water vapor near the tropopause.

Figure 4b–f also shows the horizontal extent of the monsoon anticyclone on the 395 K isentropic surface and the vertical structures of horizontal winds, meridional winds, potential temperature, and the CPT based on the five reanalyses. Most features of the time-mean upper-level anticyclone, including wind direction, wind speed, and the location of the climatological ridgeline (defined as the zero contour in zonal wind; Nützel et al., 2016), are virtually identical (see also Fig. 2). The area of the anticyclone is less consistent, as also reported by Manney et al. (2021). Defining the boundary of the anticyclone on
the 395 K potential temperature surface as the $369\,200 \, \text{m}^2 \, \text{s}^{-2}$ MSF contour, the area enclosed by this boundary is smallest in CAMS. JRA-3Q and ERA5 are in good agreement with each other, while MERRA-2 and M2-SCREAM show substantially larger anticyclones at this level. The larger anticyclones in MERRA-2 and M2-SCREAM relative to other reanalyses are well known (Manney et al., 2021; Tegtmeier et al., 2022), and may result in part from a sharper decline in total diabatic heating near the convective cloud top (primarily due to cloud radiative effects), which implies a stronger source of anticyclonic vorticity.
Warm biases in the MERRA-2 upper troposphere (Fig. 2c) may also contribute, as these biases essentially deform isentropic surfaces and deepen the air column between the upper troposphere (∼350 K) and lower stratosphere (∼400 K).

Figure 5 shows the vertical and horizontal structure of ozone anomalies relative to the zonal mean based on Aura MLS and the five reanalyses (see also Figure S4 for mean distributions of absolute mixing ratios). Large volumes of negative values in Fig. 5 illustrate the regional dilution of ozone in the ASM UTLS by deep monsoon convection (i.e. the 'ozone valley'; Bian

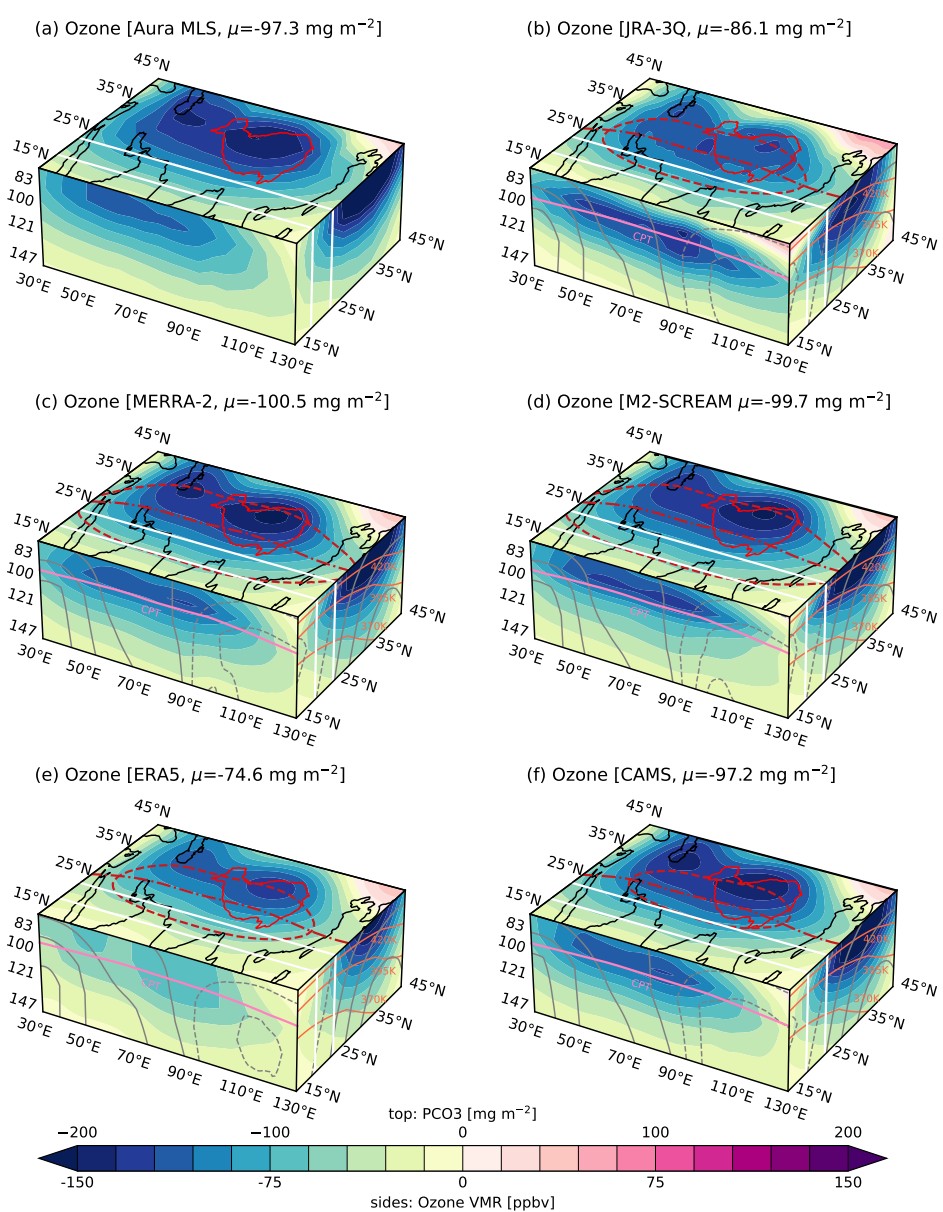

**Figure 5.** As in Fig. 4, but for ozone. Area-mean values of partial column ozone anomalies relative to the zonal mean are shown in each panel title.

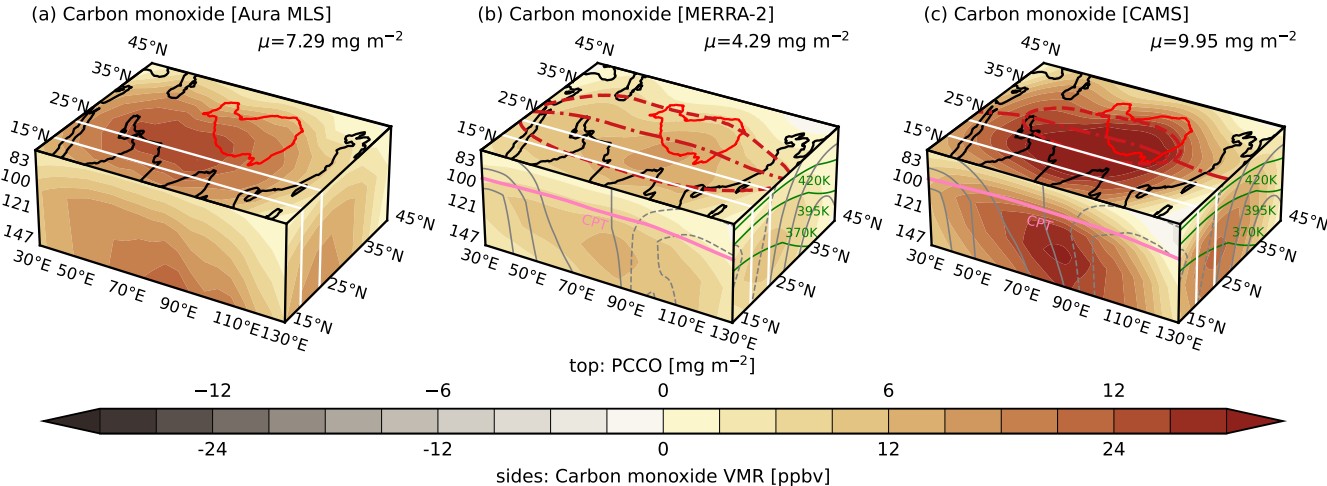

**Figure 6.** As in Fig. 4, but for carbon monoxide (CO). Area-mean values of partial column CO anomalies relative to the zonal mean are shown in each panel title.

et al., 2011). The ozone minimum is located north and west of the water vapor maximum (Fig. 4), with the largest negative anomalies in the lower stratosphere ($\sim$68–83 hPa). All five reanalysis products reproduce the horizontal distribution of the ozone valley, but with the maximum anomalies typically located slightly lower in altitude than those observed by Aura MLS ($\sim$83–100 hPa). This downward shift is particularly pronounced along the southern flank of the anticyclone in JRA-3Q, which assimilates only total column ozone (Table 2). Meanwhile, ERA5 underestimates the amplitude of the regional anomaly by nearly 25%. JRA-3Q and ERA5 also tend to overestimate ozone in the UTLS on the hemispheric scale (Fig. S4), with biases relative to Aura MLS of +28% and +15%, respectively. Positive biases in absolute PCO3 in the monsoon UTLS (Fig. S4) are small in M2-SCREAM (+1%), MERRA-2 (+4%) and CAMS (+5%), indicating that good agreement in the regional anomaly (Fig. 5) extends to the hemispheric mean in these reanalysis systems.

Figure 6 shows vertical and horizontal distributions of regional anomalies in CO relative to the zonal mean for Aura MLS, MERRA-2, and CAMS (see Fig. S5 for mean distributions of absolute volume mixing ratios). Positive anomalies of CO in the monsoon UTLS relative to the zonal mean provide an additional indication of the sources and transport of convective detrainment, which increases CO (Fig. 6) while diluting ozone (Fig. 5). These three tracers are therefore highly complementary (see also Gottschaldt et al., 2018; von Hobe et al., 2021). Water vapor mixing ratios are simultaneously affected by both convective transport and slow dehydration due to cold temperatures (see budget decompositions in sect. 4), while high values of CO and low values of ozone indicate a stronger influence of deep convection.

Both MERRA-2 and CAMS capture the main characteristics of regional anomalies in CO (i.e. the vertical and horizontal locations and extent of CO maxima) during 1 May–2 October, despite discrepancies in the magnitude of CO mixing ratios relative to Aura MLS (Fig. 6, Fig. S5). Absolute biases are especially large in MERRA-2 (–46%; Fig. S5). CAMS overestimates the

regional anomaly (Fig. 6c) and the absolute mixing ratio (Fig. S5c) by almost the same amount ($\sim$2.6 ppbv; +5%), indicating that the small high bias relative to Aura MLS is primarily regional in scope. The small magnitude of CO and large absolute bias in MERRA-2 (Fig. S5b) reflect the provisional representation of CO in this system and the accidental omission of some biofuel related sources (K. Wargan, personal communication). However, the ability to qualitatively reproduce the regional anomaly (Fig. 6b) is encouraging and suggests that even this provisional implementation of CO can be a useful tracer of transport and convective sources to the UTLS. CO and simplified CO-like tracers are often used in aerosol and chemistry-climate models as indicators of transport pathways and timescales (Shindell et al., 2008; Orbe et al., 2018; Pan et al., 2022). The direct inclusion of such tracers in reanalyses, as in the case of MERRA-2, would be valuable for both reanalysis intercomparison and model evaluation.

### 3.3 Seasonal cycles of UTLS trace gases

Figure 7 shows the mean seasonal evolutions of water vapor, ozone, and CO based on Aura MLS and those reanalysis products that provide them in pentads from 1 May through 2 October. All reanalysis products roughly capture the mean seasonal cycles of water vapor, ozone, and CO, but with offsets relative to Aura MLS. M2-SCREAM, which assimilates Aura MLS retrievals of both species and includes a stratosphere-focused chemistry scheme, unsurprisingly provides the closest match to Aura MLS in water vapor and ozone. CAMS provides an excellent representation of CO, especially at levels below the tropopause.

The upward spiraling 'tape recorder' of water vapor in the monsoon UTLS is evident in the seasonal cycles at different levels. Water vapor concentrations peak in July at 147 hPa and August at 100 hPa, but grow continuously throughout the warm season at 68 hPa (left column of Fig. 7). The amplitude of the water vapor seasonal cycle at 147 hPa (Fig. 7g) is larger in CAMS than in Aura MLS, presumably due to convective moistening simulated by the model during the peak monsoon season (July–August). The amplitude of the seasonal cycle in MERRA-2 and ERA5 is comparable to that in Aura MLS, with positive biases of 5 ppmv in ERA5 and 10 ppmv in MERRA-2 throughout the season, while M2-SCREAM has a slight negative bias in the peak monsoon season and a slightly weaker seasonal cycle (Fig. 7g). At 100 hPa, increases in water vapor in May are delayed in JRA-3Q and MERRA-2 (Fig. 7d). After June, JRA-3Q follows the seasonal evolution of water vapor at this level with a negative offset of about 1 ppmv, suggesting that the delay imposes a persistent dry bias that lasts through the entire monsoon season (Fig. 7d). A closer look at the spatial distributions of water vapor during May–September between Aura MLS and JRA-3Q (Fig. S6) indicates substantial qualitative mismatches in the global distribution of water vapor at pressures less than 100 hPa. Similar issues in stratospheric water vapor in JRA-55 (e.g. Davis et al., 2017) have been attributed to assimilation increments in tropospheric water vapor influencing water vapor non-locally (Kosaka et al., 2024). Unlike ERA5 (which disallows increments in water vapor for $p < 100$ hPa) or MERRA-2 (which relaxes stratospheric water vapor to a climatology), JRA-3Q allows assimilated observations to influence water vapor through the whole depth of the model. Although the detrimental impacts of this approach on stratospheric water vapor are reduced in JRA-3Q relative to JRA-55, the qualitative mismatch remains.

Peak water vapor concentrations at 100 hPa are larger and occur earlier in MERRA-2, ERA5, and CAMS (Fig. 7d), suggesting that these systems may overestimate the role of direct convective injection near this level. It is also possible that Aura MLS misses transient moistening at these altitudes due to its broader spatial footprints, deeper vertical weighting functions, or

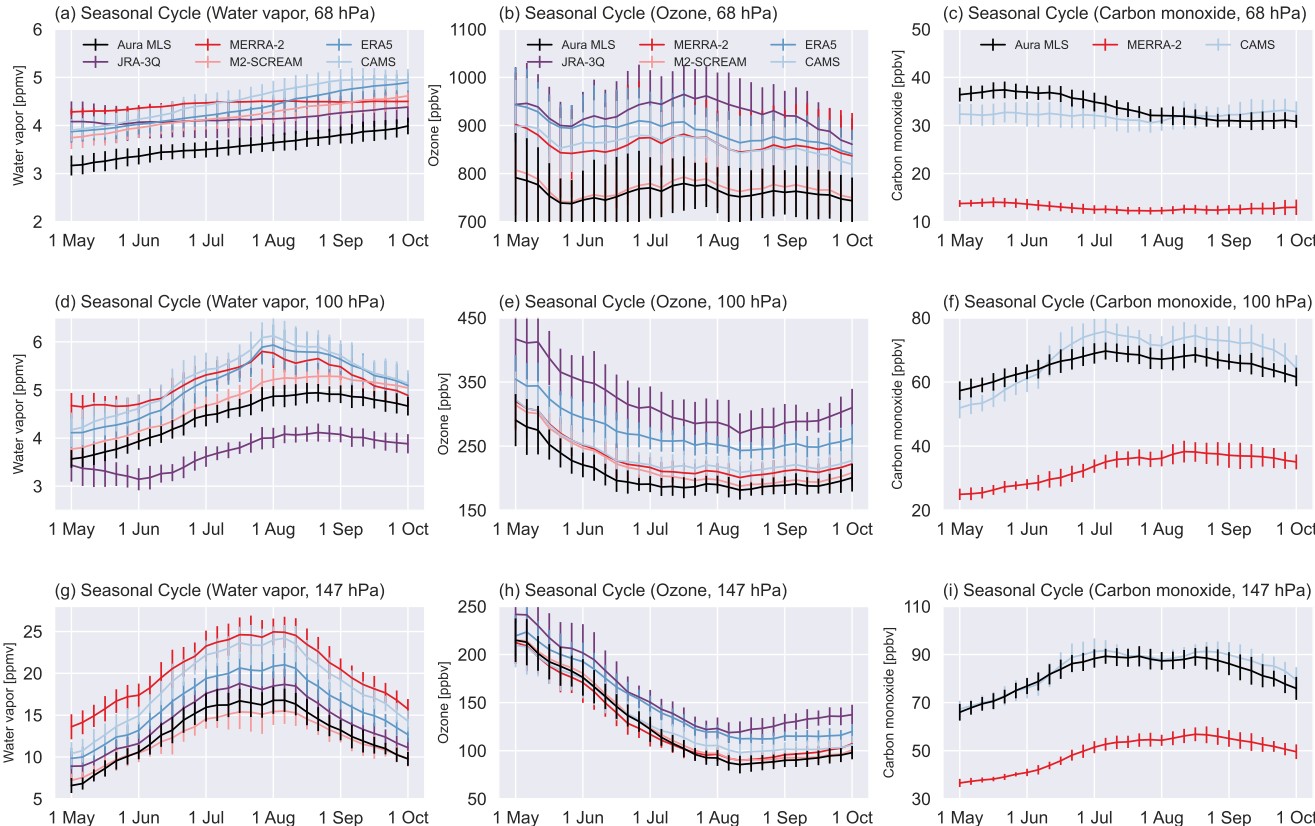

**Figure 7.** Mean seasonal evolution of area-mean (left column) water vapor, (center column) ozone, and (right column) carbon monoxide (CO) volume mixing ratios averaged over 30°E–130°E and 15°N–45°N on the (a)–(c) 68 hPa, (d)–(f) 100 hPa, and (g)–(i) 147 hPa pressure levels at 5-day intervals from 1 May through 2 October. Water vapor and ozone time series are shown for Aura MLS, JRA-3Q, MERRA-2, M2-SCREAM, ERA5, and CAMS, while CO time series are shown for Aura MLS, MERRA-2, and CAMS only. Short vertical lines in each time series mark the interannual standard deviation for each pentad.

diurnal sampling schedule. However, the lack of any corresponding signal in the Aura MLS observations suggests that episodic moistening by very deep convection does not exert a persistent effect on area-mean water vapor concentrations at 100 hPa. All

five reanalysis products overestimate water vapor relative to Aura MLS at 68 hPa (Fig. 7a). MERRA-2 and JRA-3Q show flatter seasonal cycles than Aura MLS, while the other reanalyses produce seasonal cycles with similar amplitudes and a positive bias of ∼1 ppmv (Fig. 7a). MERRA-2 strongly relaxes stratospheric water vapor to a climatology, thus damping variability above the tropopause (Davis et al., 2017). The more humid regional UTLS on 1 May in MERRA-2 relative to the other reanalyses may be inherited in part from this climatology, especially at higher levels (Fig. 7a,d,g). Error bars representing interannual

variability in MERRA-2 are noticeably smaller than those based on other reanalyses.

The reanalysis products show good qualitative agreement with Aura MLS with respect to the ozone seasonal cycle, although most systems show systematic positive biases at all levels (Fig. 7, center column). The largest ozone concentrations based on Aura MLS and the reanalyses are found in the pre-monsoon, followed by extended periods of lower ozone concentrations at 147 hPa and 100 hPa with minima in early August (Fig. 7e,h). The seasonal cycle at 68 hPa is different, with a local minimum in late May followed by a temporary increase through the peak monsoon and slow decline as the monsoon retreats (Fig. 7b). These variations in the lower stratosphere are most pronounced in JRA-3Q. JRA-3Q and ERA5 overestimate ozone through the full depth of the monsoon tropopause layer (Fig. 7b,e,h; see also Fig. 5 and Fig. S4), while MERRA-2 and CAMS are in good agreement with Aura MLS at 147 hPa and 100 hPa but overestimate ozone relative to Aura MLS at 68 hPa.

Both MERRA-2 and CAMS capture the observed evolution of the mean seasonal cycle of CO at 147 hPa and 100 hPa (Fig. 7, right column). Key differences include the amplitude of the seasonal cycle, which is more pronounced in CAMS at 100 hPa (Fig. 7f) and the large negative bias in MERRA-2 relative to Aura MLS and CAMS as mentioned above (Fig. 7c,f,i). The small seasonal variations seen in Aura MLS CO retrievals at 68 hPa are essentially absent in both reanalyses (Fig. 7c). Despite the large negative bias, MERRA-2 produces similar seasonal variations of CO at 147 hPa and 100 hPa (Fig. 7f,i), again supporting the argument for implementing CO or simplified CO-like tracers in reanalysis systems.

## 4   Tendency budgets for water vapor and ozone

In this section, we further evaluate reanalysis representations of water vapor and ozone distributions in the ASM tropopause layer by exploring the processes that shape those distributions in recent meteorological reanalyses. Three types of processes are considered as outlined in Sect. 2.1: parameterized physical and chemical processes (e.g. cloud microphysics, turbulent mixing, ozone photochemistry), dynamics (tracer flux divergence), and data assimilation. The physics term (denoted $S_{\mathrm{phy}}$) comprises the net effect of all physical parameterizations in the forecast model used to generate the background state for the reanalysis. For ozone, we combine this term with the source–sink term due to ozone chemistry as represented in the reanalysis models (Table 2). The dynamics term ($S_{\mathrm{dyn}}$) corresponds to the effects of advection and divergence. In the case of MERRA-2 this term is provided with the reanalysis, but in the cases of JRA-3Q and ERA5 we have computed it using winds and tracer distributions from the final reanalysis state after data assimilation. The assimilation or analysis increment ($S_{\mathrm{asm}}$) represents the effects of data assimilation in the tracer budget. Again MERRA-2 provides this term directly, while assimilation increments for ERA5 and JRA-3Q must be computed by comparing forecast (background state) and analysis (after data assimilation) tracer distributions. JRA-3Q only provides ozone after data assimilation, so we cannot compute the ozone assimilation increment in this reanalysis. Combining these three sets of tendencies with the local time rate of change (e.g. $\partial q/\partial t$) in the constituent budget equation (1), we retrieve a 'diffusive' residual term. We interpret this residual as primarily representing the transport due to high-frequency or small spatial scale motions resolved by the reanalysis model but not by our calculation of $S_{\mathrm{dyn}}$, together with the effects of numerical diffusion.

Figure 8 shows net dynamical and physical water vapor tendencies. Distributions are shown for the 83 hPa isobaric surface (the closest level to the cold point tropopause; Fig. 3), the east–west transect along the south flank of the anticyclone (20°N–

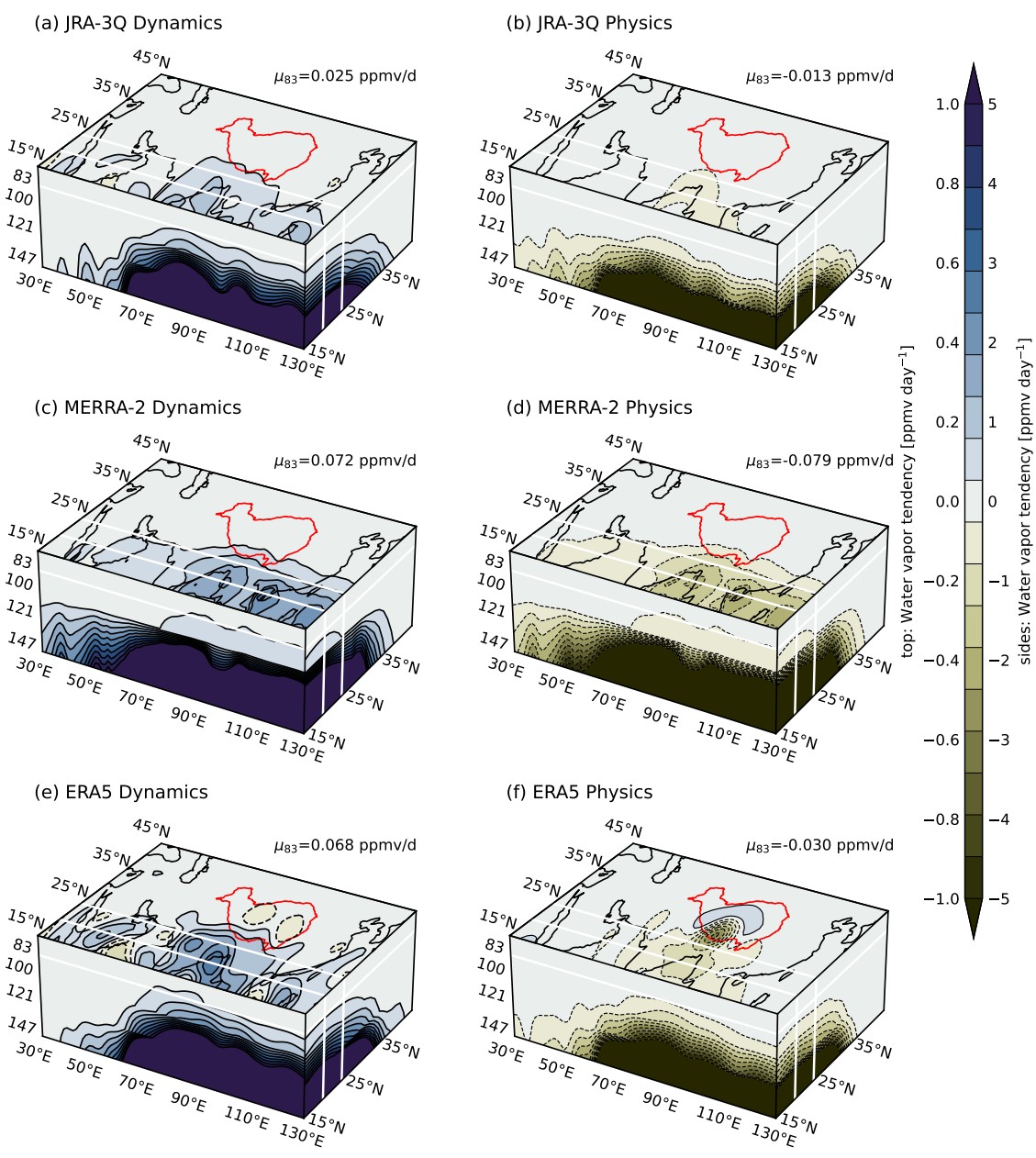

**Figure 8.** Water vapor tendencies from (left) dynamics based on analysis fields and (right) parameterized physics based on forecasts for (a,b) JRA-3Q, (c,d) MERRA-2, and (e,f) ERA5. Tendencies are shown for the 83 hPa isobaric surface (top; area-mean values listed in upper right), the east–west transect averaged meridionally over the 20°N–25°N latitude range (left/south side), and the zonal average over the 30°E–130°E longitude range (right/east side) averaged over May–September 2005–2021. Red contours mark the location of the Tibetan Plateau. White lines mark the boundaries of the east–west transect and the vertical location of 83 hPa (i.e. the top face).

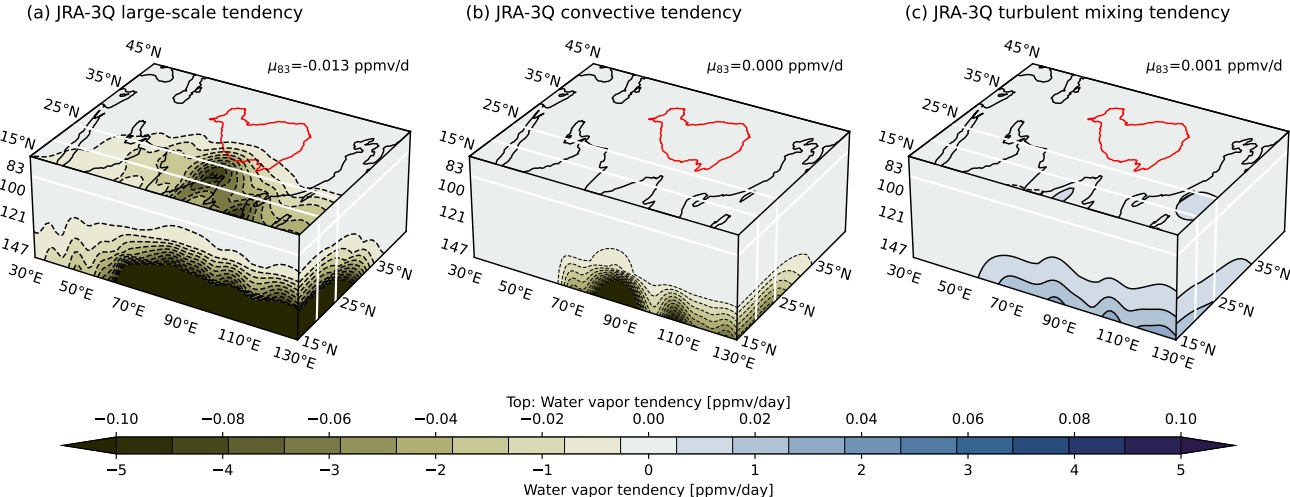

**Figure 9.** Water vapor tendencies comprising the physics term in JRA-3Q, including tendencies from parameterized (a) large-scale cloud and precipitation processes, (b) convection, and (c) vertical turbulent diffusion. Red contours mark the location of the Tibetan Plateau. White lines mark the boundaries of the east–west transect (20°N and 25°N) and top face (83 hPa; area-mean values listed in upper right), as in Fig. 8.

25°N), and the north–south distribution of the zonal average across the monsoon region (30°E–130°E). The three reanalyses
agree in the broad features of these distributions. In particular, the primary balance in all three reanalyses is between dynamical moistening and physical drying (i.e. 'advection–condensation'; Liu et al., 2010, and references therein), with the largest tendencies in the southeastern quadrant (Fig. 8). The detailed decomposition of the physics term provided by JRA-3Q indicates that the negative tendencies due to physics are dominated by large-scale condensation (Fig. 9; see Fig. S7 in the online supplement for a version of this figure showing distributions at 121 hPa rather than 83 hPa). This drying is centered in the cold trap
region over the Bay of Bengal, where all three reanalyses show local maxima in the (negative) total physics tendency (Fig. 8). MERRA-2 shows a secondary maximum in the far southeast over the South China Sea (Fig. 8d), while ERA5 shows a striking circular maximum over the southern Tibetan Plateau (Fig. 8f). The origins of the sharp peak in ERA5 are unclear. Neither MERRA-2 nor ERA5 provides a detailed breakdown of the moistening rate due to parameterized physics.

Although MERRA-2 shows the strongest dynamical moistening among the three reanalyses, it also produces the strongest
drying tendency due to model physics. Net (three-dimensional) moisture flux convergence also covers a broader region in MERRA-2, extending further to the north and west compared to JRA-3Q and ERA5 and featuring a sharper 'cutoff' between 121 hPa and 100 hPa. Dynamical moistening at 83 hPa is concentrated over the Bay of Bengal for JRA-3Q and ERA5 (Fig. 8a,e). By contrast, MERRA-2 produces larger tendencies over the South China Sea (Fig. 8c). This larger dynamical moistening in the UTLS above the South China Sea is in turn offset by larger physics-induced drying there (Fig. 8d). MERRA-2
produces much smaller values of OLR over the South China Sea and western North Pacific (Fig. S8), indicating greater deep

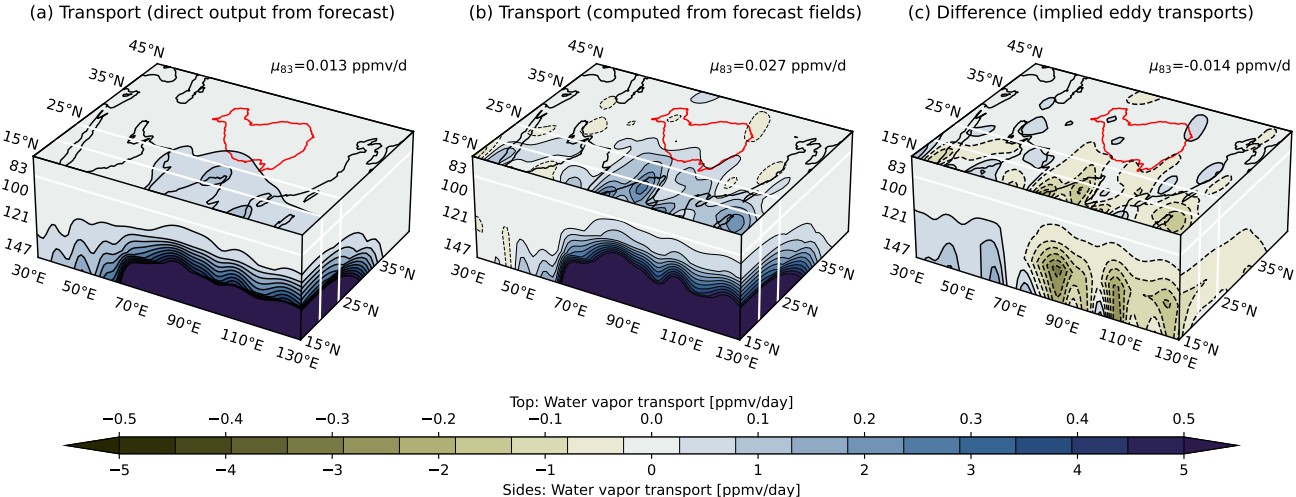

**Figure 10.** Water vapor dynamics terms based on JRA-3Q from (a) direct forecast outputs (the 'adiabatic moistening rate' provided by the reanalysis), (b) net moisture flux convergence computed from forecast fields based on eq. (1), and (c) the difference between (a) and (b), taken to represent $S_{\text{res}}$ in eq. (2). Red contours mark the location of the Tibetan Plateau. White lines mark the boundaries of the east–west transect (20°N and 25°N) and the vertical location of the distribution shown on the top face (83 hPa; area-mean values listed in upper right).

convective activity over this region in MERRA-2 than in JRA-3Q or ERA5. Comparison with observed OLR from CERES EBAF indicates that MERRA-2 overestimates convective activity in this region.

The smallest area-mean dynamical tendency on 83 hPa is that based on JRA-3Q (0.025 ppmv day$^{-1}$; Fig. 8a), while ERA5 and MERRA-2 produce similar net tendencies at 0.068 ppmv day$^{-1}$ and 0.072 ppmv day$^{-1}$, respectively (Fig. 8c,e). However,
we should revisit some distinctions in how these terms are computed (see sect. 2.1 for details). Whereas MERRA-2 provides dynamical water vapor tendencies as direct outputs from the reanalysis 'corrector' step (Gelaro et al., 2017; Fujiwara et al., 2017), we compute these tendencies based on analysis fields for JRA-3Q and ERA5. Although the latter approach affords more flexibility, it also introduces some uncertainties in how terms are assigned and budget closure.

To better illustrate the issue, Figure 10 shows an additional set of dynamical tendencies based on JRA-3Q. In addition to a
detailed breakdown of the physics terms (Fig. 9), JRA-3Q provides the diagnosed moistening rate due to dynamics over each 6-hour step of the forecast. This term (Fig. 10a), named the 'adiabatic moistening rate' or 'admr' in JRA-3Q, is aggregated at the native model spatio-temporal resolution (∼40 km on 12-minute time steps). By contrast, our calculations (Fig. 10b) are based on 6-hourly three-dimensional wind and specific humidity fields on a 1.25°×1.25° pressure-level grid. The difference between the adiabatic moistening rate and our computation applied to JRA-3Q forecast fields is shown in Fig. 10c. We interpret
this difference as the net effects of transports that are resolved by the forecast model but not by our moisture flux divergence calculation; i.e. $S_{\text{res}}$ in eq. (2). This term is mainly negative, especially in the southeastern quadrant, and offsets roughly 50% of the dynamical moistening from large-scale resolved transports at 83 hPa. The magnitudes of the resolved dynamical terms

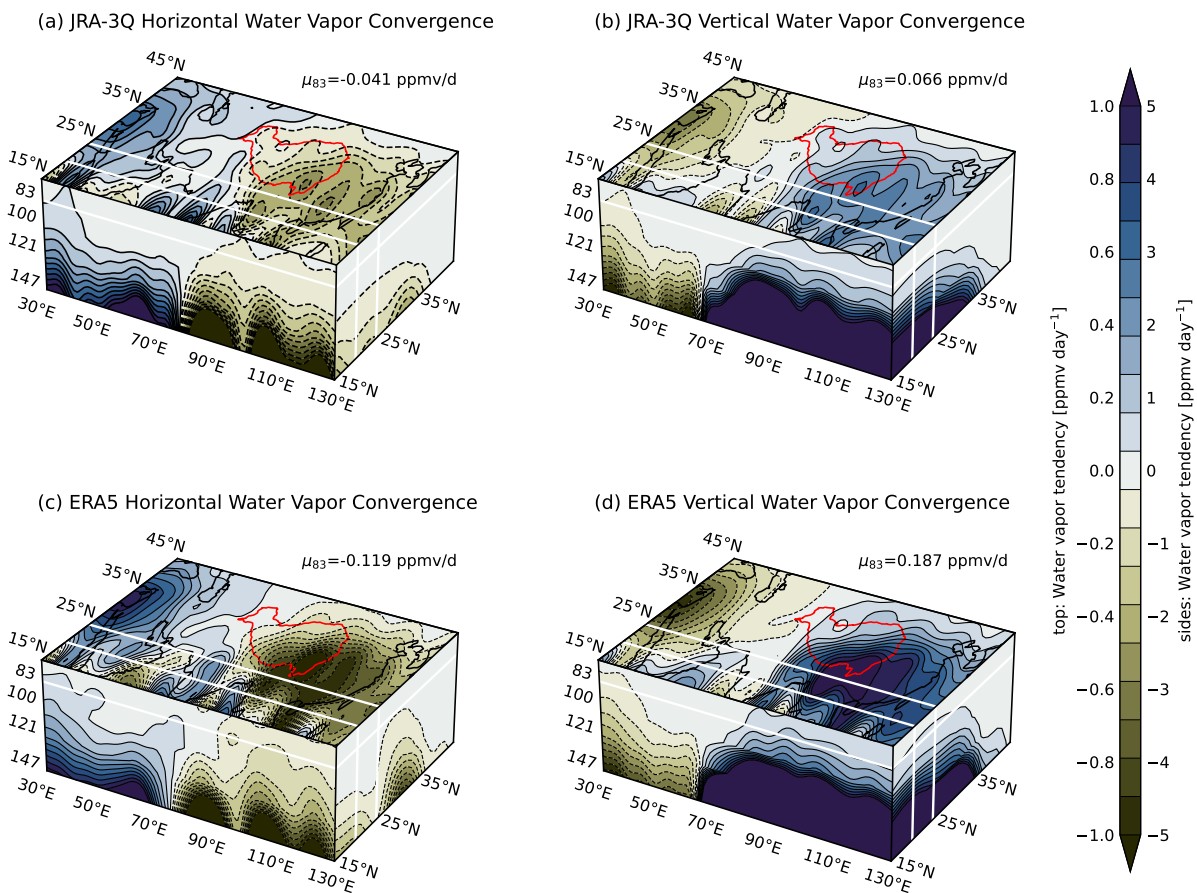

**Figure 11.** Water vapor tendencies due to (a) horizontal moisture flux convergence in JRA-3Q, (b) vertical moisture flux convergence in JRA-3Q, (c) horizontal moisture flux convergence in ERA5, and (d) vertical moisture flux convergence in ERA5. Red contours mark the location of the Tibetan Plateau and white lines mark the boundaries of the east–west transect (20°N and 25°N) and top face (83 hPa; area-mean values listed in upper right), as in Fig. 8.

on the 83 hPa isobaric surface are roughly double those for physics in both JRA-3Q and ERA5 (Fig. 8). Fig. 10 suggests that most of this difference arises from the unresolved $S_{\mathrm{res}}$ term. Dynamical moistening is thus very likely to be smaller than indicated by our calculations for JRA-3Q and ERA5. If we assume that $S_{\mathrm{res}}$ consistently offsets ∼50% of the resolved dynamical moistening, the magnitude of net dynamical moistening including $S_{\mathrm{res}}$ is roughly twice as large in ERA5 as in JRA-3Q and twice again as large in MERRA-2 as in ERA5. Strong qualitative agreement among the reanalyses is thus at least partially undermined by quantitative discrepancies, even though these discrepancies largely cancel out when the physics and dynamics terms are combined.

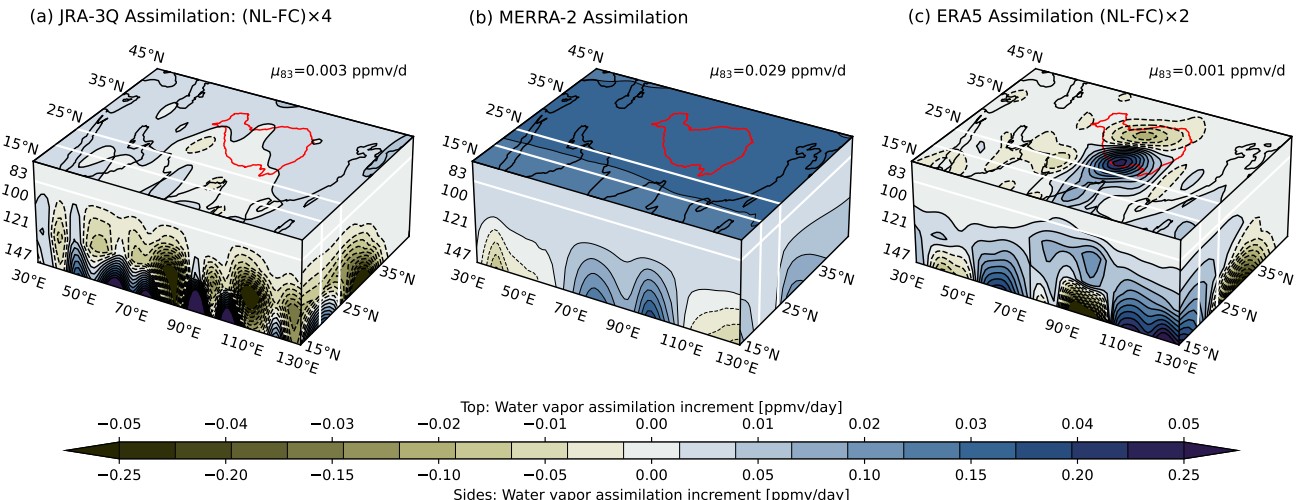

**Figure 12.** Assimilation increments for (a) JRA-3Q, (b) MERRA-2, and (c) ERA5. Red contours mark the location of the Tibetan Plateau. White lines mark the boundaries of the east–west transect (20°N and 25°N) and top face (83 hPa; area-mean values listed in upper right), as in Fig. 8.

The mismatch in magnitudes extends beyond the net dynamics and physics terms to individual components. Figure 11 shows the horizontal (left) and vertical (right) moisture flux convergence terms from JRA-3Q (upper row) and ERA5 (lower row). Although qualitatively similar, both moisture flux convergence terms based on ERA5 are between two and three times larger than those based on JRA-3Q. The ratio of area-mean horizontal moisture flux divergence to vertical flux convergence on the 83 hPa isobaric surface is consistent, however, at 0.62 for JRA-3Q and 0.63 for ERA5. The positive anomaly in time-mean water vapor relative to the zonal mean in the southeastern quadrant is maintained by vertical moisture flux convergence, consistent with both vertical moisture divergence ($q(\partial\omega/\partial p)$) and vertical moisture advection ($\omega(\partial q/\partial p)$) being negative on average. Referring to Fig. 4, vertical moisture divergence is negative in the southeastern quadrant of the anticyclone because $q$ is positive definite and $\partial\omega/\partial p < 0$ above the strong deep convection in this region. Vertical moisture advection is negative because vertical motion is upward toward lower pressures ($\omega < 0$) and $q$ decreases upward (Fig. 7). The reanalyses indicate that moisture supply to the western part of the anticyclone is primarily provided by horizontal moisture flux convergence. In the southwest, moisture flux convergence results primarily from easterly moisture advection (the prevailing winds are from the east and $q$ increases eastward). However, this transport is throttled by the cold trap above the Bay of Bengal. Horizontal convergence also contributes to the positive horizontal moisture flux convergence in the southwest, as tropical easterlies approach the western turning point around the anticyclone. In the northwest, net moisture flux convergence results from westerly flow along the northern flank of the anticyclone exporting drier air eastward and southerly flow importing more humid air from the south.

Figure 12 shows time-mean assimilation increments for JRA-3Q, MERRA-2, and ERA5. Whereas assimilation increments for MERRA-2 are provided directly with the reanalysis, assimilation increments for JRA-3Q and ERA5 are computed as

average differences between analysis and forecast specific humidities multiplied by the number of assimilation cycles per day (sect. 2.1). Data assimilation increments are an order of magnitude smaller than the physics or dynamics terms (Fig. 8).

However, because the physics and dynamics terms largely offset each other, assimilation terms of these magnitudes are large enough to 'tip' the balance. For reference, Fig. 7 indicates net changes over the full season of roughly 1 ppmv at 100 hPa and 68 hPa (moistening rates at 83 hPa are similar; Fig. S9 in the online supplement). The scale of these changes suggests an average net moistening rate of about $0.006$ ppmv day$^{-1}$, on the same order of magnitude as the assimilation increments calculated for JRA-3Q and ERA5 and one order of magnitude smaller than the area-mean increment in MERRA-2. Accordingly, despite their

smaller magnitudes, assimilation increments exert important influences on the water vapor budget at these altitudes. JRA-3Q, MERRA-2, and ERA5 adopt considerably different approaches to data assimilation and its impacts on water vapor at these altitudes. In the following paragraphs, we treat each in turn, from the most constrictive approach to the most permissive.

MERRA-2 applies the most constrictive approach, as it nudges stratospheric water vapor to a climatology. This nudging eliminates virtually all meaningful variability in water vapor at pressures less than 50 hPa and damps variations considerably

down to the tropopause. Direct observations of water vapor are also only assimilated at pressures greater than 300 hPa (Davis et al., 2017), although assimilated satellite radiances and radio occultation bending angles may influence water vapor fields at higher altitudes. Moreover, the globally integrated analysis increment in water vapor is constrained to be zero in MERRA-2 (Gelaro et al., 2017). This constraint aids mass conservation while still allowing increments to be large locally (Takacs et al., 2016), as seen for the ASM tropopause layer in Fig. 12b. Increments in MERRA-2 are uniformly positive on the 83 hPa

surface, where they are largely offset by the nudging term (Fig. S9 in the online supplement). Increments at lower altitudes show adjustments that increase moisture over the Bay of Bengal and East Asia while decreasing moisture over the Arabian peninsula and South China Sea. Although we cannot directly attribute increments to particular sources of error without targeted data assimilation experiments, these increments are consistent with corrections for (1) suppressed convective depth over the core monsoon regions due to warm bias-induced stabilization of the upper troposphere (Fig. 2) and (2) exaggerated convective

activity over the South China Sea (Fig. S8). The large positive assimilation increments in MERRA-2 along the eastern flank of the anticyclone may also compensate for 'leakage' of the negative nudging term from the extratropical lower stratosphere (Fig. S9).

ERA5 assimilates water vapor observations up to 100 hPa and then suppresses increments above this level by setting the vertical correlations of background errors in water vapor to zero. This approach is akin to assuming a perfect model at these

470 altitudes, motivated not by an expectation that the model is actually perfect but rather that the assimilated observations do not provide useful information. However, the constraint applies only to water vapor and not to other assimilated variables. ERA5 uses an incremental 4-dimensional variational (4D-Var; see Wright et al., 2022, their sect. 2.3) data assimilation system in which the entire forecast trajectory is iteratively adjusted to optimize agreement between the forecast and the assimilated observations. Accordingly, water vapor in the stratosphere is still affected by increments in winds or temperatures, as seen in

Fig. 12c. The most prominent feature in the assimilation increment is the north–south dipole that straddles the Tibetan Plateau, which evidently compensates for physics-induced tendencies of the opposite sign (Fig. 8f).

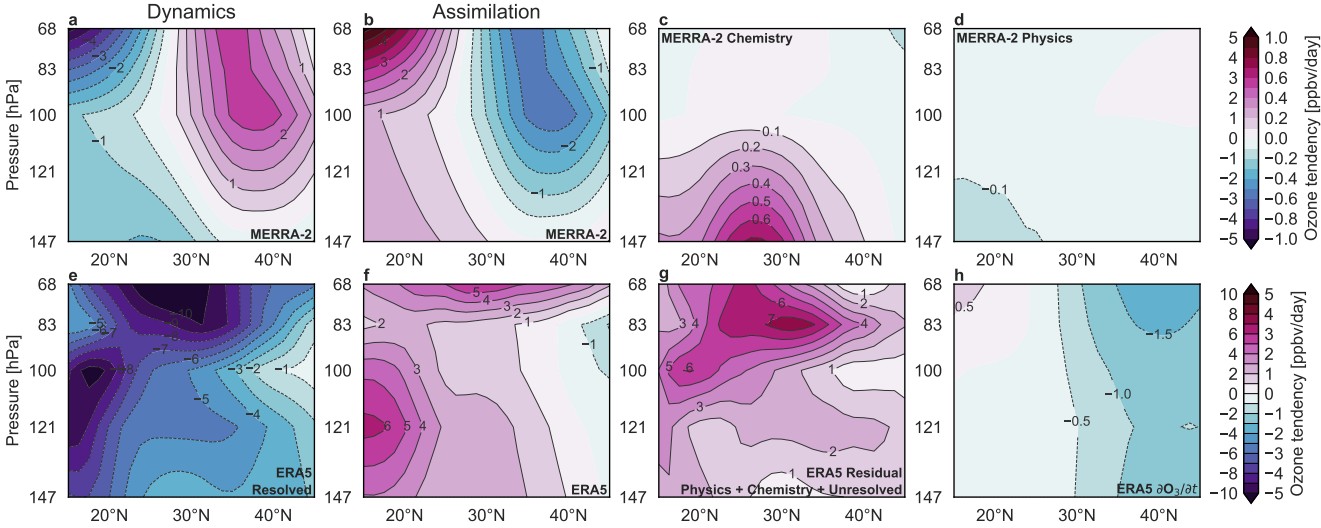

**Figure 13.** Ozone tendencies in MERRA-2 due to (a) dynamics, (b) data assimilation, (c) chemistry, and (d) parameterized physics and ozone tendencies in ERA5 due to (e) dynamics, (f) data assimilation, (g) all other processes (including chemistry, physics, and transports not resolved by our calculation of advection), and (h) average net changes over the monsoon season. Zonal averages are computed over 30°E–130°E for May–September 2005–2021. Tendencies shown in (a)–(d) are provided directly as diagnostics from the MERRA-2 predictor–corrector cycle, while those shown in (e)–(h) are computed using model-level fields from the ERA5 forecast and analysis state.

JRA-3Q allows assimilated satellite radiances and radio occultation bending angles to influence water vapor at all altitudes and does not apply a nudging to stratospheric water vapor. Radiosondes also have no strict altitude restriction, but radiosonde humidity measurements collected at temperatures below $-40$°C are not used. However, as discussed by Kosaka et al. (2024), assimilation increments in tropospheric water vapor may influence specific humidities at and above the tropopause. Figure 12a shows that assimilation increments on the 83 hPa isobaric surface are negative along much of the southern flank of the anticyclone, where JRA-3Q has a dry bias relative to Aura MLS in the lower stratosphere (Fig. S6 in the online supplement). Conversely, increments are positive along the entire northern flank of the anticyclone, where JRA-3Q has a moist bias relative to Aura MLS. Although these biases effectively offset each other in the regional mean (Fig. S10), they represent a substantial redistribution of water vapor within the monsoon lower stratosphere, from the southeastern quadrant of the anticyclone to the northern flank of the anticyclone. Moreover, even though the rates are small, the relatively long lifetime of water vapor in the stratosphere allows their effects to build up. The physics and dynamics terms in JRA-3Q show good qualitative consistency with the other reanalyses. We therefore speculate that the distribution of lower stratospheric water vapor could be substantially improved in this region by adopting an approach similar to that used in ERA5: suppress direct assimilation influences on water vapor at pressures less than 100 hPa by setting background errors to zero.

Figure 13 presents a breakdown of the ozone budget based on MERRA-2 and ERA5. MERRA-2 is the only reanalysis we evaluate here that provides these terms directly, while ERA5 provides model-level forecast and analysis fields from which

we can compute some of the terms. In contrast to water vapor, the primary balance in the MERRA-2 ozone budget is between dynamics and data assimilation (Fig. 13a,b). Contributions from physical parameterizations and chemistry are roughly an order of magnitude smaller than those from dynamics and data assimilation (Fig. 13c,d). The reliance on data assimilation to maintain the ASM ozone valley in MERRA-2 highlights the limitations of current ozone treatments in meteorological reanalyses. Most meteorological reanalyses use linearized ozone models with prescribed (usually zonal mean) production and loss rates (e.g. Cariolle and Teyssèdre, 2007; Davis et al., 2017). For example, MERRA-2 uses the 'PCHEM' scheme described by Nielsen et al. (2017) with monthly zonal-mean ozone production and loss rates taken from two-dimensional chemical transport model simulations.

Unlike MERRA-2, ERA5 does not provide diagnostic ozone tendencies due to physics or chemistry. As a result, these terms are lumped together with the residual ($S_\mathrm{res}$). Among the terms shown in Fig. 13e-h, only the assimilation increment (Fig. 13f) and the net time rate of change in ozone (Fig. 13h) are well constrained. However, these calculations are sufficient to establish that the assimilation increment contributes at leading order to the ozone budget in ERA5 as well. Remarkably, assimilation in ERA5 acts to reinforce rather than reduce high biases in ozone relative to Aura MLS. As discussed above, both regional (Fig. 5) and zonal-mean (Fig. S4) biases are positive in ERA5, resulting in ERA5 overestimating area-mean PCO3 by $\sim$60 mg m$^{-2}$ ($\sim$14%). The amplification of positive biases by data assimilation is particularly pronounced along the southern flank of the anticyclone (Fig. 13f).

These findings underscore the need for a deeper investigation into the mechanisms governing ozone in the UTLS in composition-focused reanalyses with more sophisticated chemistry, like CAMS. Companion datasets to the composition reanalyses that assimilate meteorological data but not ozone would also be useful for assessing the added value of improving the ozone models used in meteorological reanalyses. Such products would be an especially valuable comparison dataset for 'specified dynamics' simulations performed by nudging chemistry–climate models to meteorological reanalyses (Orbe et al., 2020), and could be conducted within existing frameworks such as the Chemistry–Climate Modeling Initiative (e.g. Morgenstern et al., 2017).

## 5 Summary and outlook

Reanalysis water vapor, ozone, and CO products generally compare well with Aura MLS observations in their representations of regional anomalies specific to the monsoon UTLS. This agreement is unsurprising for ozone because most reanalyses assimilate earlier versions of Aura MLS. In addition to representing the regional ozone anomalies well, MERRA-2, M2-SCREAM, and CAMS are in good agreement with Aura MLS in absolute concentrations. By contrast, ERA5 (which assimilates Aura MLS) and JRA-3Q (which does not) exhibit positive biases in absolute ozone (Fig. 7) and are unable to fully reproduce the ozone valley (Fig. 5). The agreement in CO is pleasantly surprising given concerns about the usability of the MERRA-2 product in particular (MERRA-2 omitted certain emissions sectors for CO; K. Wargan, personal communication). We recommend that more reanalyses include CO or CO-like transport tracers, which would provide valuable context for chemistry–climate and aerosol model assessments even without assimilation of CO measurements.

The consistency in regional anomalies is perhaps most surprising for water vapor, which has traditionally been shunned as unreliable in the UTLS. Although all reanalysis products exhibit moist biases in the ASM UTLS relative to Aura MLS (see also Davis et al., 2017; Krüger et al., 2022), these biases are primarily hemispheric in scale. The ability of the reanalyses to reproduce the regional anomalies well thus indicates that reanalysis water vapor products may be more reliable in active regional conduits like the monsoon anticyclone than in the global stratosphere where biases can build up over time. Among the five reanalyses evaluated here, M2-SCREAM (which assimilates Aura MLS water vapor) and JRA-3Q (which does not) are in best agreement with Aura MLS for partial column water vapor in the UTLS (68–147 hPa). However, the good agreement in JRA-3Q masks compensation between dry biases near the tropopause (83–100 hPa) and moist biases in the upper troposphere (121–147 hPa) and lower stratosphere (68 hPa) relative to Aura MLS (Fig. 7; Fig. S6). The dry bias near the tropopause in JRA-3Q is initially perplexing because this reanalysis produces the warmest cold point temperatures, but may be explained by vertical 'imprints' of assimilation increments from lower levels (Kosaka et al., 2024, see also Fig. S6). A similar issue affected the earlier JRA-55 reanalysis (Davis et al., 2017; Fujiwara et al., 2022). Although reduced tropospheric moisture increments have reduced the magnitude of biases in JRA-3Q (Kosaka et al., 2024), our results indicate that the issue remains.

Detailed analysis of the water vapor budget near the tropopause shows that JRA-3Q, MERRA-2, and ERA5 all consistently reproduce the dominant 'advection–condensation' balance expected for this region (e.g. Fueglistaler et al., 2005; Schoeberl et al., 2012), despite differences in the magnitudes of the individual terms (Figs. 8–9). Differences in the primary budget terms reflect differences in convective sources, moisture gradients, resolved wind speeds, and model physics. Moreover, despite the lack of strong observational constraints, assimilation increments are of comparable magnitude to the net advection–condensation balance (Fig. 12). Data assimilation remains influential even when direct increments are suppressed in ERA5, as increments in winds and temperatures can still influence the analyzed water vapor distribution.

Data assimilation exerts an even greater influence on ozone. MERRA-2, M2-SCREAM, and CAMS, all of which assimilate Aura MLS ozone over this period, reliably reproduce the observed 'ozone valley', a regional-scale dilution of tropopause-layer ozone above the ASM relative to the zonal mean (Fig. 5). However, the MERRA-2 ozone budget indicates that this feature is maintained primarily by data assimilation, with ozone tendencies due to physics and chemistry an order of magnitude smaller than the advection and data assimilation terms (Fig 13). Despite assimilating Aura MLS ozone, ERA5 plainly underestimates the amplitude of the ozone valley (see also Tegtmeier et al., 2022, their Figs. 8.62–8.63), with assimilation appearing to reinforce rather than reduce this bias (Fig 13). Meanwhile, the ozone valley is located at a lower altitude in JRA-3Q, possibly because JRA-3Q only assimilates total column ozone observations, not ozone profiles (Kosaka et al., 2024).

Aura MLS is set to be retired in 2026. Given this impending data gap, it would be immensely valuable to have a broader set of reanalyses that produce physically meaningful variations of water vapor at and above the tropopause. Previous work has suggested that only the ECMWF reanalyses, and especially ERA5, provide stratospheric water vapor products that meet these criteria (e.g. Davis et al., 2017; Wang et al., 2020). To this shortlist we can certainly add M2-SCREAM and other MLS-centered reanalyses (e.g. Errera et al., 2019), with the caveat that the future availability and utility of these products will be curtailed by the impending retirement of Aura MLS. However, assimilation of water vapor retrievals in the stratosphere is not the only route to improving stratospheric water vapor. Our results lead us to suggest that the relative utility of stratospheric water vapor

in ECMWF reanalyses derives not from better resolving the cold point (other recent reanalyses perform comparably well in this aspect; Tegtmeier et al., 2020), better representations of vertical motion (ascent in the tropical stratosphere has tended to be too fast in ECMWF reanalyses; Wright et al., 2011; Tao et al., 2019; Tegtmeier et al., 2022), the inclusion of methane oxidation (Untch et al., 1998), or other model-specific features. Instead, we argue that ECMWF reanalyses produce physically meaningful variations in stratospheric water vapor because they suppress direct data assimilation increments in water vapor above the tropopause by setting background errors to zero. This approach, which essentially gives the model control of anomaly evolution above the tropopause, aligns well with the advection–condensation paradigm for stratospheric water vapor (e.g. Liu et al., 2010). Its success thus depends on two criteria. First, the vertical resolution around the tropopause should be fine enough to resolve the cold point, a requirement well satisfied by the current generation of atmospheric reanalyses (Tegtmeier et al., 2020). Second, the vertical discretization around the tropopause and the advection scheme should be sufficient to inhibit spurious vertical diffusion across the tropopause (e.g. Hardiman et al., 2015). The second criterion could be tested by evaluating residence times within the tropopause layer based on transport tracers (e.g. CO). The calculated transit times can then be compared to expected transit times inferred from diabatic heating (Gettelman et al., 2010; Tegtmeier et al., 2022), supplemented where possible by comparisons with aircraft measurements (e.g. Krüger et al., 2022).

Knowland et al. (2025) recently demonstrated an alternative approach using the CoDAS data assimilation system used by M2-SCREAM. Their results show that even sparse stratospheric water vapor measurements could improve the fidelity of stratospheric water vapor in reanalyses. Their experiments used measurements from the Stratospheric Aerosol and Gas Experiment (SAGE) III instrument. SAGE III provides only 15–30 profiles per day globally, but these measurements are of high quality in the stratosphere (Davis et al., 2021). Combined with the long lifetime of water vapor in the stratosphere, just a few high-quality measurements per day are sufficient to shift the analysis state significantly. Unaffected by assimilation of any other observations, the model then propagates these adjusted states forward. Although SAGE III retrievals are not currently provided on the near-real-time schedule required by meteorological reanalyses, the results are nonetheless promising.

The experiments conducted by Knowland et al. (2025) used a data assimilation system that assimilates retrieved quantities directly. Experiments in more comprehensive assimilation systems would be required to ensure that the benefits derived by assimilating SAGE III or similar measurements would not be overwhelmed by imprints of humidity increments due to satellite radiances with peak sensitivity at lower levels. This possibility is a particular concern because the SAGE III data is subject to periodic interruptions. Accordingly, both the zero-background-error approach adopted by ECMWF and the sparse-assimilation approach proposed by Knowland et al. (2025) depend on suppressing spurious increments due to the tails of assimilated radiance weighting functions. Both methods also depend on a good representation of stratospheric water vapor in the model used to generate the background state.

Either approach would benefit from the inclusion of CO or CO-like transport tracers, which would support additional diagnostics targeting convective source distributions, ascent rates, and mixing processes along the dynamical tropopause. For example, the inclusion of CO would help to identify active troposphere-to-stratosphere conduit regions in each reanalysis by distinguishing air recently lofted to levels at and above the tropopause. Only two of the reanalyses we evaluate provide estimates of CO (CAMS and MERRA-2). Although CAMS does not assimilate CO retrievals from Aura MLS, the two datasets agree

extremely well. Moreover, although CO concentrations in MERRA-2 are much smaller than observed, the regional anomalies and mean seasonal cycle in the ASM tropopause layer are still in good agreement with Aura MLS. These results highlight the potential utility of CO as a transport tracer in reanalysis products, even in a rudimentary unassimilated form. Inclusion of CO would further support process-oriented intercomparison and analysis of atmospheric transport in environments like the monsoon, where boundary layer pollution and convective transport play pivotal roles (Pan et al., 2022; Qie et al., 2025).

*Data availability.* Aura MLS (Lambert et al., 2021; Schwartz et al., 2021a, b, c), MERRA-2 (GMAO, 2015a, b, c, d), and M2-SCREAM (GMAO, 2022) data were acquired from the NASA Goddard Earth Sciences Data and Information Services Center (GES DISC; https://disc.gsfc.nasa.gov). CERES EBAF (NASA/LARC/SD/ASDC, 2023) data were acquired from the Atmospheric Science Data Center (ASDC) operated by NASA Langley (https://ceres-tool.larc.nasa.gov). ERA5 products (Hersbach et al., 2017, 2023a, b) are available from the Copernicus Climate Data Store (CDS; https://cds.climate.copernicus.eu), while CAMS reanalysis products are distributed through the Copernicus Atmosphere Data Store (ADS; https://ads.atmosphere.copernicus.eu/cdsapp#!/dataset/cams-global-reanalysis-eac4). JRA-3Q reanalysis products (JMA, 2022) were acquired from the Data Integration and Analysis System (DIAS; https://search.diasjp.net/en/dataset/JRA3Q) archive maintained by the Japan Agency for Marine-Earth Science and Technology (JAMSTEC) and the University of Tokyo.

*Author contributions.* JSW, SZ, and JC conceived the study, wrote the initial draft of the paper, and conducted the analysis. SMD, PK, ML, XY, and GJZ provided feedback on the analysis and assisted with interpretation. GJZ, JSW, and PK acquired funding for the work. All authors contributed to writing and revising the manuscript.

*Competing interests.* One of the authors is a guest coordinator for the "SPARC Reanalysis Intercomparison Project (S-RIP) Phase 2" special issue in Atmospheric Chemistry and Physics. The authors have no other competing interests to declare.

*Acknowledgements.* We thank Dr Jianchun Bian for valuable comments on an earlier version of the analysis, Dr Kris Wargan for assistance confirming details of the MERRA-2 and M2-SCREAM reanalyses, Dr Yayoi Harada for assistance confirming details of the JRA-3Q reanalysis, and Dr Hans Hersbach, Dr Antje Inness, and Dr Peter Bechtold for assistance confirming details of the ERA5 and CAMS reanalyses. This work has been supported by the National Natural Science Foundation of China (grant number 42275053), the Beijing Natural Science Foundation (grant number IS23121), and the Hong Kong Research Grants Council (project no.16300424 ).

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
