# Peer review of "Online supplement to “Evaluating reanalysis representations of climatological trace gas distributions in the Asian monsoon tropopause layer”"

_EGUsphere, 2025_

## Referee Comment (RC2)

**Review of "Evaluating reanalysis representations of climatological trace gas distributions in the Asian monsoon tropopause layer " by Wright et al.**
* * *
**Summary**

The aim of this work is to assess the performance of five different modern meteorological reanalyses in terms of their ability to reproduce the characteristic seasonal variations in UTLS trace gas composition (H2O, O3, CO) associated with the Asian summer monsoon. The core of this work is an intercomparison of H2O, O3 and CO distributions with AURA MLS observations in the area associated with the ASM. The second objective of the work is to relate and explain the evaluated trace gas concentrations and their variability in the models to different mechanisms (dynamics, physics, assimilation).

The analysis shows that the selected meteorological reanalyses are in principle able to represent the typical seasonal changes in H2O, O3, CO. It also shows substantial differences between individual models in the trace gas balance, which is a relevant result for the scientific community using this model output.

The topic of this work is well in line with the subject areas of ACP.

The data analysis seems adequate to me, but I have some general and specific comments that could help to improve the structure of the scientific paper and increase its comprehensibility. I recommend this work for publication, subject to revision/consideration of these comments.
* * *
**General comments**

1.

P11 Figure 3: The water vapor concentration drops across the UTLS by several orders of magnitude with highest water vapor in UT. So, I wonder whether the signal (overestimation of PCWV by all models) that you see in the top face is purely dominated by the troposphere. It would be interesting to look at the PCWV for H2O (and for the other trace gases) also in a narrowly defined layer around the tropopause level, and separately in the UT and in the LS.

2.

Chapter 1: I would appreciate to get more context on what is known on the accuracy/performance of the reanalyses with respect to these trace species in the UTLS. Are there well-known substantial H2O/Ozone/CO biases? Are these biases expected to be stronger in the ASM?

3.

Chapter 4: This chapter should be revised using a clearer structure, simpler language, by creating a logical connection to chapter 3 in order to give the reader more guidance and make it easier to follow.

In particular, more context (especially at the beginning of the chapter) on physical/dynamic tendencies would help the reader understand why you are using them (based on the findings in sect. three) and how to interpret them.

The authors could also consider adapting the chapters heading 'Water vapour and ozone budgets' and linking it directly to the tendencies.

4.

P19: Figure 9: The use of CERES EBAF data (Fig. 9) appears somewhat out of nowhere here (also in sect. 2.2, P5 ll. 145). Although this figure is interesting and supportive for your conclusions (regarding the overestimation of deep convection by MERRA-2), you can consider whether it is sufficient to append it to the supplement. Here, the OLR difference (CERES-each model) would also be interesting here to demonstrate also differences in OLR between CERES and the other models.
* * *
**Technical corrections**

P2: ll. 30-31: What exactly is meant by 'relatively'? In general, this term is used very frequently in the paper and should be specified -if possible.

P2: ll. 32-33: "*Convection also leads to relatively low concentrations of ozone and high concentrations of CO and other pollutants (…)".* Please specify which altitude region you are referring to (UT, LS, both?)

P2: ll. 41-42: "*at these altitudes".* Do you mean the UTLS in general here?

P4: ll.80-82: What is the vertical resolution/number of levels in the various models in the height range relevant for this study? This could be interesting information for the reader. A comment and/or a corresponding addition to Table 1 would be helpful in this regard.

P5: ll. 125-125: *"subtracting forecast specific humidities (before data assimilation) from analysis specific humidities (after data assimilation)".* To avoid misunderstandings, it may be useful to mention that data assimilation increments are the difference between the analysis and the background forecast (i.e., the first guess).

P5: ll 135: *"upper troposphere and stratosphere (p < 316 hPa)"* Is there also an upper boundary of MLS data in the lower stratosphere?

P5: ll. 142: *"tropopause based on the World Meteorological Organization (WMO) definition".* Please clarify that you are referring to the cold point tropopause? Could you add a reference for its WMO definition?

P6: ll. 146: Can you also comment on the vertical resolution of the different AURA MLS products? Is that different for the different trace species?

P10 ll. 210-211: Is there a reason why you chose the lapse-rate tropopause here and not the cold-point tropopause (which you use in the rest of the paper)?

P10 ll 219: *"Although the reanalysis profiles are considerably warmer than indicated by Aura MLS…"* Please rephrase, for example: Although the reanalysis temperature profiles are considerably warmer than indicated by Aura MLS at all altitudes within the UTLS…

In the following cold bias in MLS is emphasized only. It should be also mentioned, that reanalyses are also affected by biases in the UTLS region. For ECMWF IFS (ERA5 based on IFS cycle 41r2), for example, there is a well-known warm bias at the tropopause (e.g., Ingleby 2016; Ingleby et al., 2017), and a cold bias in the LS radiatively which is associated with a collocated moist bias. Ingleby (2016) indicates that the warm tropopause may to some extent be caused by insufficient vertical resolution.

P11 Figure 3 (GENERAL COMMENT): The water vapor concentration drops across the UTLS by several orders of magnitude with highest water vapor in UT. So, I wonder whether the signal (overestimation of PCWV by all models) that you see in the top face is purely tropospheric. It would be interesting to look at the PCWV for $H_2O$ (and the other trace gases in other figures) also in a narrowly defined layer around the tropopause level, and separately in the UT and in the LS.

P12 ll. 245-246: "*The small biases in the regional anomalies based on these reanalyses establish that moist biases relative to Aura MLS are hemispheric in scale and are not specific to the monsoon region*" What exactly do you mean by that?

P14 ll. 285-286: In my point of view, the agreement between CAMS and Aura MLS CO distributions is pretty strong, while there is a substantial offset in MERRA-2 CO. Hence, I would recommend to make these differences a bit more prominent.

P15 ll. 300: I would recommend to delete "*unsurprisingly*".

P15 ll. 302: "*at lower levels*" → change to: at all levels

P15 ll. 305: What do you mean by "*changes*"? Differences between the minimum in May and maximum in Aug.? Please rephrase.

P15 ll. 305: "*Changes in water vapor at 147 hPa (Fig. 6g) are larger in CAMS and JRA-3Q*" → Looking at Fig.6g, I guess you mean MERRA-2 instead of JRA-3Q? Please clarify and revise the full sentence.

In the following up to ll. 309 (P16) it is difficult the follow the description, in particular what figure (panel) you are referring to. I therefore cannot trace the stated biases (5/10 ppm) in ERA5/MERRA-2 given in ll.308.

P16 ll.309: "early part" →maybe change to: In May

P16 ll.331-332: "good qualitative agreement" followed by "persistently high distortions" sounds a little contradictory when written in the same sentence.

P16 ll. 332: "*largest ozone concentrations*" → largest ozone concentrations in Aura MLS and the reanalyses

P19 Figure 9 (In general comments): The use of CERES EBAF data (Fig. 9) appears somewhat out of nowhere here (also in sect. 2.2, P5 ll. 145). Although this figure is interesting and supportive for your conclusions (regarding the overestimation of deep convection by MERRA-2), you can consider whether it is sufficient to append it to the supplement. Here, the OLR difference (CERES-each model) would also be interesting here to demonstrate also differences in OLR between CERES and the other models.

P24 ll. 458 and following: description of Figure 13 is not quite clear to me and could be improved. I would recommend to rephrase it.

P25 ll. 470-471: *"Remarkably, assimilation in ERA5 acts to reinforce rather than reduce high biases in ozone relative to Aura MLS (see Fig. 4, Fig. 6, and Fig. S4 in the online supplement)".* This result is interesting indeed. However, I see this reinforcement only in Fig. 6, but according to Fig.4 (e) the increments seem to act in the right way (at least positive). Can you comment on this? Maybe P. 26 ll. 509 should be also revised then, in case you decide to change text on P25.

P25 ll. 481: *"Reanalysis water vapor, ozone, and CO products generally compare well with Aura MLS observations, especially in their representations of regional anomalies specific to the monsoon"*

I think this sentence could be improved to do justice to the results of the paper. I agree that the basic regional trace gas distributions are reproduced quite well, but you also found that there are also major differences especially in the seasonal variability between the different models and between model and MLS.

The sentence after ll. 482-483 reads a bit confusing. More explanations are necessary why this is a surprising/unsurprising result (maybe by means of literature references).

P25 ll. 485. Do you mean the moist bias in the lower stratosphere, or which altitude region? I would also recommend to replace the reference Krüger et al., (2022) by e.g., Davis et al. 2017 who show indications for a moist bias in the LS in different reanalyses.

P26 ll.503: "*Data assimilation exerts an even greater control on ozon*" control → influence

P26 ll. 514-515: "*However, assimilation of water vapor retrievals in the stratosphere is not the only route to improving stratospheric water vapor".*

I fully agree that assimilation of humidity is not the only key to improve water vapor in the stratosphere. Data assimilation is not a tool to remove systematic features – but by nature a data assimilation ideally should act to reduce them. You can also use data assimilation diagnostics to learn a lot about the mechanisms (e.g., dynamics) causing systematic biases.

In P26 II. 529-531 you provide strong conclusions/recommendations how the assimilation of observations in the stratosphere should be treated in the models.

In my opinion, this recommendation is probably too strong, as it has not been proven in your study whether Aura MLS is exclusively responsible for the assimilation increments.

This could be analyzed by means of an Observing System Experiment (data denial) which allows to attribute increments to the assimilation of particular observation types (e.g., Aura MLS). This is certainly beyond the scope of your work, but maybe you should consider narrowing your statement. Can you comment on this?

---

## Author Response (AR1)

**1 Response to reviewer 1**

We thank the reviewer for their careful consideration of the manuscript. Our responses to all comments and questions are provided below. Line numbers refer to the updated manuscript without tracked changes.

> **Comment 1.1**
>
> To look into the seasonal evolution of tracers in the UTLS from prior to monsoon to post monsoon you have considered the period from May to October. However, majority of the studies mainly focussed on the chemical composition within ASMA between July and August corresponding to peak monsoon months (or June to September). Why did you consider the zonal anomalies of the PCWV, PCO3 and PCCO from May to October.

**Response:**

Thank you for this question. While most previous studies have focused on the peak monsoon months, adopting May–September as the analysis period enables a more comprehensive examination of the seasonal cycle and subseasonal variability of composition in this region. The selected period is motivated by results from two recent studies examining this region, namely those of Santee et al. (2017) based on MLS observations and Manney et al. (2021) based on reanalysis products.

Although the anticyclone is strongest during the peak monsoon months, anomalies in the thermodynamic structure and composition of the monsoon UTLS are often initiated in May as pre-monsoon convection intensifies and typically extend to the end of September when the anticyclone dissipates (see, e.g., Santee et al. 2017, their Fig. 3, and Manney et al. 2021, their Fig. 2). Our analysis period ends on 2 October (the pentad centered on 30 September), consistent with MLS observations for the 370 K and 390 K potential temperature surfaces (Santee et al. 2017, their Fig. 2).

In our work, the extended analysis period allows us to better explore how the pre-monsoon state affects concentrations during the peak monsoon season, as discussed in Section 3.3 (persistent dry bias at 100 hPa in JRA-3Q, which develops from a small bias at the beginning of May) and our companion paper (persistent moist or dry anomalies associated with interannual variability; Zhang et al. 2025).

> **Comment 1.2**
>
> There is a shift in the bias (both for O3 and WV) in all the reanalysis compared to MLS which has limitations due to the coarser vertical resolution. Is the vertical resolution in all the reanalysis used in this study are nearly same?

**Response:**

The underlying vertical resolutions of these reanalyses are different but all are at least as fine as Aura MLS (Fig. R1; added to the manuscript as Fig. 1). For the intercomparison, all five reanalysis datasets were interpolated to consistent pressure levels of 68 hPa, 83 hPa, 100 hPa, 121 hPa, and 147 hPa that match those of Aura MLS water vapor and ozone (CO retrievals and comparisons adopt the coarser spacing of 68 hPa, 100 hPa, and 147 hPa).

For ERA5, CAMS, MERRA-2, and M2-SCREAM, we have retrieved model-level fields with finer vertical resolution (see Fig. R1). We interpolated these model-level fields to pressure levels directly, accounting for spatio-temporal variations in surface pressure (Sections 2.1 and 2.3). For JRA-3Q, we have used pressure–level fields. Whereas the standard pressure-level grids of other reanalyses do not include a level between 70 hPa and 100 hPa, JRA-3Q includes a pressure level at 85 hPa. The vertical spacing of pressure levels from JRA-3Q is thus already consistent with that of Aura MLS.

Vertical biases relative to Aura MLS may arise in part from the interpolation, but sensitivity tests examining distributions before and after interpolation indicate that such effects are small, consistent with these reanalyses typically having native vertical levels close to the MLS levels (Fig. R1).

In addition, as mentioned in the text, we have not applied the MLS vertical weighting functions in this analysis. Here, our main interest is in reanalysis intercomparison and the potential for reanalyses to help close the impending gap in observations of this region. The vertical interpolation step thus serves primarily to establish consistent weights for the vertical integrals and for convenience in creating the plots.

[Figure]

Figure R1: Vertical levels for a surface pressure of 1000 hPa for the five reanalyses examined in the manuscript. Both pressure levels and model levels are shown for JRA-3Q; we used the pressure-level fields in this case. Other reanalyses provide pressure-level products on the standard pressure-level grid (far left), which is insufficient to match the 83 hPa Aura MLS level. The pressure levels used in the analysis are marked by horizontal dotted lines.

**Comment 1.3**

For better clarity in the interest of readers, it would be useful if you mention in the figure caption what the values in the square brackets represent, though you have mentioned in the body of the manuscript.

**Response:**

Thank you for this suggestion. We have added explanations of these values to all captions in the revised manuscript.

**Comment 1.4**

What could be the possible reasons for the negative and positive tendencies in the water vapor over the southern and northern part of the Tibetan Plateau? Which dominate (physics, advection, assimilation)?

**Response:**

This pattern (as seen in Fig. 7f and Fig. 12c of the manuscript) is dominated by the physics term (Fig. 7f), largely offset by the assimilation (Fig. 12c). In our calculation, these two terms are computed independently: the physics term from the model moisture tendency due to parameterized physics and the assimilation increment from the difference between forecast and analysis fields. Unfortunately, we cannot pinpoint the source more exactly than that based on the published outputs, as ERA5 only provides a total physics tendency. We have followed up with contacts at ECMWF. They were unaware of the issue but speculated that it may be related to gravity wave activity and/or breaking around the southern fringes of the Tibetan Plateau (P. Bechtold; personal communication 8 April 2025), although turbulent mixing and condensation (i.e. all of the relevant parameterization) may also contribute. If a more detailed explanation becomes available after this manuscript is published, we will provide it as a corrigendum.

**Comment 1.5**

Lastly, I would suggest to write many simpler sentences when much information has to be conveyed so that it would be easy for the readers.

**Response:**

We appreciate your suggestion and have revised the text accordingly.

**2  Response to reviewer 2**

We thank the reviewer for their careful consideration of the manuscript. Our responses to all comments and questions are provided below.

**2.1  General comments**

> **Comment 2.1**
>
> P11 Figure 3: The water vapor concentration drops across the UTLS by several orders of magnitude with highest water vapor in UT. So, I wonder whether the signal (overestimation of PCWV by all models) that you see in the top face is purely dominated by the troposphere. It would be interesting to look at the PCWV for H2O (and for the other trace gases) also in a narrowly defined layer around the tropopause level, and separately in the UT and in the LS.

**Response:**

Thank you for raising this question. Overestimates are not dominated solely by the upper troposphere. As shown in Figure 6 of the manuscript, the reanalyses systematically overestimate water vapor in the lower stratosphere by 20–60% throughout the vertical range, with the noted exceptions of M2-SCREAM at 100–147 hPa (good agreement) and JRA-3Q (dry bias) at 100 hPa.

> **Comment 2.2**
>
> Chapter 1: I would appreciate to get more context on what is known on the accuracy/performance of the reanalyses with respect to these trace species in the UTLS. Are there well-known substantial H2O/Ozone/CO biases? Are these biases expected to be stronger in the ASM?

**Response:**

We have added several sentences in the introduction to address this suggestion (L47-L50). A more detailed response follows.

Reanalysis ozone fields have been extensively examined and evaluated for the UTLS, as reviewed by Davis et al. 2017 and Chapter 4 of the S-RIP Final Report. Chapter 8 of the S-RIP Final Report also provided an intercomparison of reanalysis ozone fields within the Asian monsoon 'ozone valley.' [*Disclaimer: S. Davis and J. Wright contributed to all three of these references*].

Overall, reanalyses can capture variations in total ozone well and feature good representations of vertical structure in the stratosphere, but with significant inter-reanalysis differences in the UTLS. These evaluations are summarized in the upper rows of Fig. R2, reproduced from Chapter 4 of the S-RIP Final Report. The recommendation of that report regarding ozone was stated as "Users should generally use caution when using reanalysis ozone fields for scientific studies and should check that their results are not reanalysis-dependent."

Reanalysis water vapor fields have been largely shunned as unreliable based on the poor performance of early reanalyses (particularly NCEP-NCAR), significant discrepancies, and the knowledge that assimilated observational constraints were so weak as to be essentially non-existent if not actively detrimental (see response to final technical comment below). Continued reservations about the quality of reanalysis water vapor products is evident in the lower rows of Fig. R2 and in the summary recommendation: "Reanalysis stratospheric water vapour fields should generally not be used for scientific data analysis (except perhaps for ERA5). Any examination of these fields must account for their inherent limitations and uncertainties."

However, these and other assessments have focused almost exclusively on meteorological reanalyses (e.g. MERRA-2 and ERA5) for which composition is a secondary objective. The ability of composition-focused reanalyses such as CAMS and M2-SCREAM to reproduce variability in this region has not been evaluated. Whereas M2-SCREAM is a stratosphere-focused product and thus expected to reproduce water vapor and ozone well around the tropopause, CAMS is troposphere-focused. In addition, JRA-3Q was released after the S-RIP Final Report and has yet to be evaluated in this context. Given the poor performance of the earlier JRA-55 with respect to stratospheric water vapor (Fig. R2; see also Davis et al. 2017), improving this aspect of the reanalysis was a stated point of emphasis in the development of JRA-3Q (Kosaka et al. 2024, their Section 9.2).

[Figure]

Figure R2: Summary assessment of reanalysis ozone and water vapor products in the zonal-mean UTLS; reproduced from Chapter 4 of the S-RIP Final Report.

A priori, we expected biases to be larger in the ASM region due to the complexity of interactions, the transition from thermodynamic to dynamical tropopause transport barriers, and the difficulty of representing convective influences. We were thus pleasantly surprised to find that the reanalyses are able to reproduce the regional anomalies well despite substantial biases in the zonal mean.
* * *
**Comment 2.3**

Chapter 4: This chapter should be revised using a clearer structure, simpler language, by creating a logical connection to chapter 3 in order to give the reader more guidance and make it easier to follow. In particular, more context (especially at the beginning of the chapter) on physical/dynamic tendencies would help the reader understand why you are using them (based on the findings in sect. three) and how to interpret them. The authors could also consider adapting the chapters heading 'Water vapour and ozone budgets' and linking it directly to the tendencies.

**Response:**

Thank you for this suggestion. We have revised the text to include more context and better connect section 4 to section 3 (L360-L375). We have also changed the section title to "Tendency budgets for water vapor and ozone"
* * *
**Comment 2.4**

P19: Figure 9: The use of CERES EBAF data (Fig. 9) appears somewhat out of nowhere here (also in sect. 2.2, P5 ll. 145). Although this figure is interesting and supportive for your conclusions (regarding the overestimation of deep convection by MERRA-2), you can consider whether it is sufficient to append it to the supplement. Here, the OLR difference (CERES-each model) would also be interesting here to demonstrate also differences in OLR between CERES and the other models.

**Response:**

Thank you for mentioning this. We were also uncertain as to whether to include the OLR distributions in the main text, and have now moved this figure to the supplement. Our primary interest in OLR in this

work is the stark difference in the location and size of the area enclosed by the $220\,\mathrm{W\,m^{-2}}$ contour. As part of the A-RIP project, we are also evaluating seasonal thermodynamic budgets in these reanalyses, in which convection plays a central role through its myriad influences on heating rates. We will defer further evaluation of OLR in these and other recent meteorological reanalyses to when we report that work.

**2.2 Technical corrections**

> P2: ll. 30-31: What exactly is meant by 'relatively'? In general, this term is used very frequently in the paper and should be specified -if possible.

In these two sentences, 'relatively' means relative to other locations at the same latitude. We have revised the second sentence to make this explicit (L32-L33). We have also reduced the use of 'relatively' through the rest of the manuscript.

> P2: ll. 32-33: "Convection also leads to relatively low concentrations of ozone and high concentrations of CO and other pollutants (...)". Please specify which altitude region you are referring to (UT, LS, both?)
> P2: ll. 41-42: "at these altitudes". Do you mean the UTLS in general here?

In both instances we mean the UTLS. We have made this explicit in the revision. (L33, L42)

> P4: ll.80-82: What is the vertical resolution/number of levels in the various models in the height range relevant for this study? This could be interesting information for the reader. A comment and/or a corresponding addition to Table 1 would be helpful in this regard.

Please see Figure R1 in the response to Reviewer 1 above. This figure has been added to the manuscript.

> P5: ll. 125-125: "subtracting forecast specific humidities (before data assimilation) from analysis specific humidities (after data assimilation)". To avoid misunderstandings, it may be useful to mention that data assimilation increments are the difference between the analysis and the background forecast (i.e., the first guess).

Thank you for suggesting this clarification. We have changed the text to "directly subtracting forecast specific humidities (based on the model background state before data assimilation) from analysis specific humidities (the reanalysis state after data assimilation)" (L132-L133)

> P5: ll 135: upper troposphere and stratosphere ($p < 316$ hPa) Is there also an upper boundary of MLS data in the lower stratosphere?

The upper bound of MLS data varies by species but is well above the stratopause for the three trace gases we evaluate in this paper. In version 5, the useful range is capped at 0.001 hPa for water vapor, ozone, and CO.

> P5: ll. 142: "tropopause based on the World Meteorological Organization (WMO) definition". Please clarify that you are referring to the cold point tropopause? Could you add a reference for its WMO definition?

We have clarified that we are referring to the lapse-rate tropopause (L155-L156). We have also added a reference to Hofmmann and Spang (2022). Although this is not the original reference (see their citation of WMO 1957), it is much easier to access and outlines the procedures to identify the tropopause defined in this way. We would be glad to add the original reference as well if needed.

> P6: ll. 146: Can you also comment on the vertical resolution of the different AURA MLS products? Is that different for the different trace species?

We have added a sentence "Whereas water vapor and ozone are provided on five levels within our analysis domain (147 hPa, 121 hPa, 100 hPa, 83 hPa, and 68 hPa; Fig. 1), CO is provided on every other level (147 hPa, 100 hPa, and 68 hPa)." (L151-L152)

> P10 ll. 210-211: Is there a reason why you chose the lapse-rate tropopause here and not the cold-point tropopause (which you use in the rest of the paper)?

We show the lapse-rate tropopause in the supplement for context, as this definition of the tropopause is more commonly used and some readers may be more familiar with it. The lapse-rate tropopause shown in the supplement is provided directly in AIRS products (we have specified "along with the distribution of lapse-rate tropopause pressure provided by AIRS" in the revision; L224), whereas we have computed the locations of the cold point tropopause. We have added a reference to Tegtmeier et al. (2020) in Section 2.1 to indicate that we have computed it following the method outlined in that paper (L111-L112).

> P10 ll 219: "Although the reanalysis profiles are considerably warmer than indicated by Aura MLS..." Please rephrase, for example: Although the reanalysis temperature profiles are considerably warmer than indicated by Aura MLS at all altitudes within the UTLS...

Revised to "considerably warmer than Aura MLS in the monsoon UTLS" (L233)

> In the following cold bias in MLS is emphasized only. It should be also mentioned, that reanalyses are also affected by biases in the UTLS region. For ECMWF IFS (ERA5 based on IFS cycle 41r2), for example, there is a well-known warm bias at the tropopause (e.g., Ingleby 2016; Ingleby et al., 2017), and a cold bias in the LS radiatively which is associated with a collocated moist bias. Ingleby (2016) indicates that the warm tropopause may to some extent be caused by insufficient vertical resolution.

We stress the cold bias in MLS relative to the radio occultation temperature profiles because ERA5 and MERRA-2 are in very good agreement with the radio occultation profiles. The radio occultation measurements have small uncertainties and agree well with high-resolution direct measurements from radiosondes (Ho et al. 2017). We have also added a sentence "JRA-3Q has a roughly 0.3 K warm bias relative to the radio occultation-based profile, ERA5, and MERRA-2" (L235-L236), which we acknowledge should have been better emphasized in the original submission.

At the vertical resolution we evaluate, the profiles from ERA5 and MERRA-2 are almost indistinguishable from the radio occultation profile (thick black line in Fig. 2a). This consistency matches the results of Tegtmeier et al. (2020), who found that ERA5 has largely eliminated the warm bias in ERA-Interim at the cold point tropopause. Tegtmeier et al. (2020) reported a warm bias in MERRA-2 at the cold point, but their results (see their Fig. 6) indicate that this warm bias is smaller in the subtropics than in the tropics.

> P11 Figure 3 (GENERAL COMMENT): The water vapor concentration drops across the UTLS by several orders of magnitude with highest water vapor in UT. So, I wonder whether the signal (overestimation of PCWV by all models) that you see in the top face is purely tropospheric. It would be interesting to look at the PCWV for H2O (and the other trace gases in other figures) also in a narrowly defined layer around the tropopause level, and separately in the UT and in the LS.

Please refer to our response to comment 2.1 above.

> P12 ll. 245-246: "The small biases in the regional anomalies based on these reanalyses establish that moist biases relative to Aura MLS are hemispheric in scale and are not specific to the monsoon region" What exactly do you mean by that?

We mean that moist biases in specific humidities in the monsoon region for each reanalysis have similar magnitudes to moist biases in the zonal mean based on the same reanalysis. This similarity indicates that the moist biases are zonally distributed and not specific to the monsoon region. We have revised the phrasing to clarify this point: "The small biases in the regional anomalies based on these reanalyses indicate that moist biases relative to Aura MLS are present in the zonal mean and are not specific to the monsoon region." (L260-L261)

> P14 ll. 285-286: In my point of view, the agreement between CAMS and Aura MLS CO distributions is pretty strong, while there is a substantial offset in MERRA-2 CO. Hence, I would recommend to make these differences a bit more prominent.

We have revised the language to further emphasize the large absolute bias in MERRA-2. (L302)

> P15 ll. 300: I would recommend to delete "unsurprisingly".

We have chosen to keep "unsurprisingly" to emphasize that the comparison of M2-SCREAM against Aura MLS is not independent. We have added "which assimilates Aura MLS retrievals of both species and includes a stratosphere-focused chemistry scheme" to clarify why this result is unsurprising. (L315-L317)

> P15 ll. 302: "at lower levels" → change to: at all levels

We are reluctant to make this change due to the significant differences in CO at 68 hPa in May and June. We have revised the sentence to "especially at levels below the tropopause". (L317)

> P15 ll. 305: What do you mean by "changes"? Differences between the minimum in May and maximum in Aug.? Please rephrase.

Yes, here we mean the amplitude of the seasonal cycle. We have revised these sentences accordingly. (L320-L324)

> P15 ll. 305: "Changes in water vapor at 147 hPa (Fig. 6g) are larger in CAMS and JRA3Q" → Looking at Fig.6g, I guess you mean MERRA-2 instead of JRA-3Q? Please clarify and revise the full sentence.

The amplitude of the seasonal cycle in MERRA-2 is comparable to that observed by Aura MLS, just offset by approximately 10 ppmv. We were referring to JRA-3Q but we agree that the larger amplitude is less pronounced in this reanalysis than in CAMS and have removed reference to JRA-3Q from this sentence. (L320-L321)

> P16 ll.309: "early part" → maybe change to: In May

Changed as suggested. (L324)

> In the following up to ll. 309 (P16) it is difficult the follow the description, in particular what figure (panel) you are referring to. I therefore cannot trace the stated biases (5/10 ppm) in ERA5/MERRA-2 given in ll.308.

We have added additional references to the figure panels throughout the discussion of Fig. 6. (L318-L357)

> P16 ll.331-332: "good qualitative agreement" followed by "persistently high distortions" sounds a little contradictory when written in the same sentence.

We have changed "persistent high biases" to "systematic positive biases". In this sentence, by "good qualitative agreement" we mean good agreement in the shape and timing of the seasonal cycle and by "systematic positive biases" we mean the offset. We believe that the distinction is clear with reference to the figure. (L346)

> P16 ll. 332: "largest ozone concentrations" → largest ozone concentrations in Aura MLS and the reanalyses

Revised to "based on Aura MLS and the reanalyses" (L346-L347)

> P19 Figure 9 (In general comments): The use of CERES EBAF data (Fig. 9) appears somewhat out of nowhere here (also in sect. 2.2, P5 ll. 145). Although this figure is interesting and supportive for your conclusions (regarding the overestimation of deep convection by MERRA-2), you can consider whether it is sufficient to append it to the supplement. Here, the OLR difference (CERES-each model) would also be interesting here to demonstrate also differences in OLR between CERES and the other models.

Please see response to comment 2.4 above. (L395)

> P24 ll. 458 and following: description of Figure 13 is not quite clear to me and could be improved. I would recommend to rephrase it.

We have revised the text to clarify the description of Figure 13. (L485-L502)

> P25 ll. 470-471: "Remarkably, assimilation in ERA5 acts to reinforce rather than reduce high biases in ozone relative to Aura MLS (see Fig. 4, Fig. 6, and Fig. S4 in the online supplement)". This result is interesting indeed. However, I see this reinforcement only in Fig. 6, but according to Fig.4 (e) the increments seem to act in the right way (at least positive). Can you comment on this? Maybe P. 26 ll. 509 should be also revised then, in case you decide to change text on P25.

Both regional (Fig. 4) and zonal-mean (Fig. S4) biases are positive in ERA5, and are thus reinforced by positive assimilation increments. The distributions in Fig. 4 are anomalies relative the zonal mean. Comparing Fig. 4e to Fig. 4a, the regional reduction in UTLS ozone is underestimated by ERA5 relative to Aura MLS (regional means: $-97\,\mathrm{mg\,m^{-2}}$ for MLS; $-75\,\mathrm{mg\,m^{-2}}$ for ERA5). Figure S4 shows that ERA5 overestimates regional-mean partial column ozone by $60\,\mathrm{mg\,m^{-2}}$, indicating that approximately 2/3 of the bias seen in Fig. 6 is in the zonal mean and 1/3 in the weaker regional anomaly. (L499-L500)

> P25 ll. 481: "Reanalysis water vapor, ozone, and CO products generally compare well with Aura MLS observations, especially in their representations of regional anomalies specific to the monsoon"
>
> I think this sentence could be improved to do justice to the results of the paper. I agree that the basic regional trace gas distributions are reproduced quite well, but you also found that there are also major differences especially in the seasonal variability between the different models and between model and MLS.

We have revised the sentence to limit its scope: "Reanalysis water vapor, ozone, and CO products generally compare well with Aura MLS observations in their representations of regional anomalies specific to the monsoon UTLS." (L511-L512) The seasonal cycle is qualitatively consistent after accounting for absolute biases. Exceptions to that consistency are noted in the following sentences.

> The sentence after ll. 482-483 reads a bit confusing. More explanations are necessary why this is a surprising/unsurprising result (maybe by means of literature references).

The explanations originally included in the parentheticals have been expanded in the revised manuscript. (L512-L521)

> P25 ll. 485. Do you mean the moist bias in the lower stratosphere, or which altitude region? I would also recommend to replace the reference Krüger et al., (2022) by e.g., Davis et al. 2017 who show indications for a moist bias in the LS in different reanalyses.

We have added a citation to Davis et al. 2017 but kept the original citation to Krüger et al. 2022, who provided additional context on the reasons behind the bias for ERA5 specifically. We have also specified that we refer to the Asian summer monsoon UTLS. (L521-L522)

> P26 ll.503: "Data assimilation exerts an even greater control on ozon" control → influence

Changed as suggested.

> P26 ll. 514-515: "However, assimilation of water vapor retrievals in the stratosphere is not the only route to improving stratospheric water vapor".
>
> I fully agree that assimilation of humidity is not the only key to improve water vapor in the stratosphere. Data assimilation is not a tool to remove systematic features – but by nature a data assimilation ideally should act to reduce them. You can also use data assimilation diagnostics to learn a lot about the mechanisms (e.g., dynamics) causing systematic biases.

We strongly agree on this point. Our motivation for constructing budget decompositions that explicitly include data assimilation is to support more effective learning about the mechanisms causing biases. However, our results support previous conclusions that data assimilation increments on stratospheric water vapor itself are currently more harmful than helpful, as they derive primarily from vertical correlations with humidity increments at lower levels. This is the case in JRA-3Q, which is much improved from JRA-55 in magnitude due to smaller tropospheric humidity increments (Kosaka et al. 2024) but still produces unrealistic distributions (our Fig. S6). MERRA-2 avoids this issue by relaxing stratospheric water vapor to a zonal-mean climatology, but at the expense of interannual and longitudinal variability.

Because ERA5 and CAMS prevent increments in stratospheric water vapor due to assimilated humidity measurements, those increments result solely from assimilation effects on temperature and winds. In this case, the assimilation increments in stratospheric water vapor are more useful because they can indicate dynamical or physical sources of biases, rather than the influences of assimilated radiances with peak sensitivity to humidity at lower altitudes.

> In P26 ll. 529-531 you provide strong conclusions/recommendations how the assimilation of observations in the stratosphere should be treated in the models.
>
> In my opinion, this recommendation is probably too strong, as it has not been proven in your study whether Aura MLS is exclusively responsible for the assimilation increments.
>
> This could be analyzed by means of an Observing System Experiment (data denial) which allows to attribute increments to the assimilation of particular observation types (e.g., Aura MLS). This is certainly beyond the scope of your work, but maybe you should consider narrowing your statement. Can you comment on this?

We apologize for the misunderstanding. Here we mean that with Aura MLS soon to retire we will lose the most reliable source of information about composition in the upper troposphere and stratosphere, not that reanalyses will lose a key source of assimilated data. The only reanalysis among the five we consider that assimilates Aura MLS water vapor is M2-SCREAM.

Of the three reanalyses, our results support previous conclusions that the ECMWF approach is best able to reproduce physically meaningful (though incomplete) stratospheric water vapor variability (Davis et al. 2017). Our reasoning is:

(1) currently assimilated observations provide few constraints on stratospheric water vapor and have been shown in several cases (such as JRA-55 and JRA-3Q; see Davis et al. 2017 or Kosaka et al. 2024) to have an adverse effect;

(2) our budget decomposition shows that the models used to generate the background state capture the dominant advection-condensation balance well; and

(3) the ECMWF reanalyses ERA5 and CAMS, which explicitly assume zero background error in water vapor above the tropopause, are the only reanalyses to produce physically meaningful space-time variability above the tropopause.

The lifetime of stratospheric water vapor is long, with entry mixing ratios set by processes near the tropopause. If the processes that set the entry mixing ratio are well resolved, the anomalies set at the tropopause can be effectively propagated forward by the model.

On OSSE experiments, another recent study provides some useful context. Knowland et al. (2025) showed that assimilating Stratospheric Aerosol and Gas Experiment (SAGE) III data from the International Space Station allowed the GEOS CoDAS (as used for M2-SCREAM but without Aura MLS) to better capture events such as the water vapor injected to high altitudes by the Hunga eruption in January 2022. SAGE III provides fewer than 30 profiles per day, indicating that forward propagation of sparse but high-quality observed anomalies is sufficient to improve assimilated representations of stratospheric water vapor.

The key to the success of the SAGE III assimilation experiment conducted by Knowland et al. (2025) is the ability to assimilate high quality measurements of stratospheric water vapor, which required adapting an existing assimilation scheme. The SAGE III measurements are also not available on the near-real-time schedule required by ongoing reanalyses. Our argument is that, in the case of stratospheric water vapor, if 'assimilatable' high quality observations are not available it would be better to ignore observations entirely. Such an approach cannot capture events such as the Hunga eruption, but it may be more palatable to reanalysis centers who require rapid data availability and/or prefer not to introduce a separate retrieval- rather than radiance-based constituent assimilation scheme just for stratospheric water vapor.

It is also worth noting that the control simulation presented by Knowland et al. (2025) is roughly analogous to the ECMWF approach, with 'replay' to MERRA-2 plus two key differences: (1) it does not relax stratospheric water vapor to a climatology and (2) it adopts the more sophisticated stratospheric chemistry module used in M2-SCREAM (StratChem; Nielsen et al. 2017), which includes a water vapor source from methane oxidation. The model is far from perfect (tropical ascent rates and methane oxidation both appear to be overestimated in the lower to middle stratosphere), but it captures interannual variability that is explicitly missing from MERRA-2.